# Role of evolving sea surface temperature modes of variability in improving seasonal precipitation forecasts
Agniv Sengupta [1] ✉, Duane E. Waliser[2,3], Michael J. DeFlorio[1], Bin Guan [2,3], Luca Delle Monache[1] & F. Martin Ralph[1]

The value of improving longer-lead precipitation forecasting in the water-stressed, semi-arid western United States cannot be overstated, especially considering the severity and frequency of droughts that have plagued the region for much of the 21st century. Seasonal prediction skill of current operational forecast systems, however, remain insufficient for decision-making purposes across a variety of applications. To address this capability gap, we develop a seasonal forecasting system that leverages the long-term memory of leading global and basin-scale modes of sea surface temperature variability. This approach focuses on characterizing and capitalizing on the spatiotemporal evolution of predictor modes over multiple antecedent seasons, instead of the customary use of predictive information from just the current season. Another distinctive methodological feature is the incorporation of sources of predictability spanning multiple timescales, from interannual to decadal-multidecadal. An evaluation of the forecast system's performance from cross-validation analyses demonstrates skill over core winter precipitation regions—California, Pacific Northwest, and the Upper Colorado River basin. The developed model exhibits superior skill compared to dynamical and statistical benchmarks in predicting winter precipitation. Experimental seasonal precipitation forecasts from the model have the potential to provide critical situational awareness guidance to stakeholders in the water resources, agriculture, and disaster preparedness communities.

Skillful and timely forecasts of precipitation are coveted by the socio-economic sectors of the semiarid western United States (U.S.). With ~75% of the total annual precipitation being received during the winter months of November through March, the water supply is primarily governed by the distribution of rainfall[1] and snowmelt[2]. The distribution, however, is far from uniform with California experiencing a uniquely high interannual variability in precipitation[3] governed by the presence or absence of a relatively small number of large storms, typically landfalling atmospheric rivers (ARs)[4,5].

Multi-year droughts have been widespread in the state during the past decade, namely from 2012 to 2016 and from 2019 to 2022, which were relieved by the unprecedented storms of winter 2022–2023[6]. Such climatic extremes are associated with great socioeconomic implications with substantial challenges for water supply and management, wildfire risks, etc. As an example, the State of California in its budget for the fiscal year 2021–2022 authorized more than $5 billion over 4 years for water resilience and drought preparedness[7]. Reliable forecasts of seasonal precipitation have the potential to provide the water management community with adequate lead time for preparing for such droughts and inform their decisions on response planning and resource positioning during extreme situations. Thus, the need for skillful forecasts beyond the weather time scales remains a critical demand for state and local water resource managers.

Forecasting initiatives that span the spectrum between weather and climate horizons have received impetus from the World Weather Research Programme/World Climate Research Programme's S2S Prediction Project[8] with applications being explored for various sectors and geographic regions, including water in the western U.S. through the Real-Time Pilot Initiative[9]. Longer lead forecasts currently available to user agencies in the western U.S. can be broadly classified into dynamical, statistical, and hybrid. Dynamical seasonal forecasting systems include the coupled models available from the North American Multi-Model Ensemble (NMME) Project[10]. Statistical forecasts of precipitation seek to

¹Center for Western Weather and Water Extremes, Scripps Institution of Oceanography, University of California San Diego, La Jolla, CA, USA. ²Jet Propulsion Laboratory, California Institute of Technology, Pasadena, CA, USA. ³Joint Institute for Regional Earth System Science and Engineering, University of California Los Angeles, Los Angeles, CA, USA. ✉e-mail: agsengupta@ucsd.edu

exploit the lagged relationships between a set of predictors and the predictand through different empirical approaches[11,12]. Additionally, there have been hybrid strategies to combine dynamical model output with empirical information[13,14]. The prediction skill of our current forecast systems, especially for precipitation, however, remains limited on seasonal timescales. While examining the seasonal forecast skill for mean precipitation and precipitation extremes, Slater et al.[15] found consistently low skill for precipitation forecasts across the western U.S. after a 2-week lead-time; Becker et al.[16] noted very little or no improvements in the seasonal forecasts of precipitation when evaluating the latest suite of NMME models including upgrades in different model versions. Such challenges in the skillful prediction of winter precipitation have manifested themselves more prominently during recent years, e.g., the dry winter of 2015–2016[17], wet winters of 2016–2017[18] and 2022–2023[6]. This quest for improvements in seasonal predictive skill using innovations in forecast methodologies serves as the primary motivation for the current study.

Climate system components with long-term memory serve as potential predictors of regional hydroclimate. For instance, sea surface temperature (SST) conditions in the global oceans have been utilized as an influential variable for predicting variations in regional precipitation[19–24]. Over the western U.S., Gershunov and Cayan[11] introduced a statistical model for predicting winter precipitation based on canonical correlation analysis (CCA) between SST forcings and observations. Alfaro et al.[25] also leveraged the CCA method for the prediction of California summer surface air temperatures. More recently, several studies have explored the role of anomalous Pacific SSTs, primarily the El Niño–Southern Oscillation (ENSO), in modulating seasonal precipitation in the U.S. West: Jong et al.[26] documented El Niño's influence on California precipitation by region and seasons, while Chapman et al.[27] explored the potential hydroclimate predictability under ENSO forcing as well as with internal variability. However, studies that seek to predict the departure from seasonal precipitation means based solely on the state of the ENSO are less effective; ENSO, in fact, explains only about 25% of the year-to-year variability in California precipitation[28,29]. This was evident during the 2015–2016 El Niño winter when the forecasted odds favoring above-normal precipitation conditions did not materialize in the U.S. Southwest and a devastating drought continued instead, while the contrary happened during the recent 2022–2023 La Niña winter. This work is therefore motivated by the need to incorporate sources of seasonal predictability beyond interannual fluctuations modulated by ENSO. It additionally considers the role of decadal-to-multidecadal sources of variability, given their influential role in modulating seasonal precipitation[30,31].

For this study, we hypothesize that characterization of the phase evolution of predictor variables, i.e., incorporation of antecedent spatiotemporal predictors from multiple seasons prior, can benefit regional precipitation predictions. Except for a few studies[12,32], there has been considerably less research attention focused on the use of longer-lead predictor information for generating better seasonal forecasts. Here we develop a *Multi-Lead Multi-Source Sea Surface Temperature* (MLMS−SST) model, illustrated in Fig. 1, to forecast seasonal winter precipitation in the western U.S. at a forecast lead of 1-season. This study differs from prior approaches in the way we utilize SST evolution leveraging an extended Empirical Orthogonal Function (extended-EOF) analysis for objective identification of modes based on concurrent consideration of spatial and temporal recurrence. This allows us to depict the spatiotemporal evolution of recurrent variability as modes (represented as a spatiotemporal sequence of patterns), unlike a canonical EOF analysis which only identifies the mature-phase spatial structure. Furthermore, in addition to the usual characterization of ENSO and related teleconnections, our methodology additionally incorporates secular warming and decadal-to-multidecadal SST variations as effective predictors of precipitation over the western U.S. Similar evolution-centric SST analyses have been employed to study the interannual variability of the East Asian monsoon[33,34] as well as for a more robust characterization of El Niño's precipitation influence on the United States and the Americas[35] relative to the widely used Niño-3.4 index, enhancing

prospects for seasonal prediction in select regions and seasons, e.g., the cold season in the western U.S. — the focus of the present study.

## Results

### Precipitation variability during winter in the western U.S

The spatial distributions of winter climatological precipitation and associated variability are shown in Fig. 2. The winter season is characterized by heavy precipitation in the western United States (Fig. 2a) with regions of precipitation maximum focused in the Pacific Northwest and northern California (seasonal mean precipitation magnitudes > 4.5 mm day$^{-1}$). The areas of precipitation maxima reflect the interaction of winter storms (predominantly ARs) with regional orography, namely the Cascades and the Sierra Nevada Mountain ranges. The Intermountain West and southern California receive much less precipitation in comparison (~1.5–2.0 mm day$^{-1}$). The interannual variability in precipitation, estimated as a measure of its standard deviation (SD), is depicted first for the entire region in winter (Fig. 2b) and then over key regions of interest in the western U.S.—northern California, southern California, and the Upper Colorado River basin (Fig. 2c–e), plotted as a function of the calendar month. The SD map highlights the large variability in year-to-year precipitation in southern California; here, the precipitation SD in winter is ~1.0–1.5 mm day$^{-1}$ whereas the climatological winter mean is ~2.0 mm day$^{-1}$. Over northern California, the SD of winter precipitation is 1.5–2.0 mm day$^{-1}$ against a climatology of ~3.5–4.0 mm day$^{-1}$. In the Upper Colorado River basin, the winter precipitation is seemingly less variable, but not when viewed relative to its regional climatology; SD values here are in the order of ~0.5 mm day$^{-1}$ whereas the background winter climatological precipitation is ~0.5−1.0 mm day$^{-1}$. Also evident in Fig. 2c, d is the significance of the November-March extended winter season (period of focus in our current study) in the U.S. West, accounting for ~75% of the annual precipitation in northern and southern California.

The observed time series of winter precipitation in the historical record over northern (bordered in red, Fig. 2) and southern California (bordered in blue, Fig. 2) is displayed in Fig. 3. The precipitation anomaly time series is constructed and depicted (as blue bars) in terms of the deviation (in percentage) from its long-term mean. The time series of precipitation is also temporally smoothed using the LOESS filter and shown as the green shaded curve. The dashed violet lines represent ±1 standard deviation in observed precipitation anomalies over the respective regions. Visually apparent from Fig. 3 are the prominent wet winters in the past few decades, namely, water year (WY) 1983 with 76% and 109% above normal over northern and southern California, WY1995 with 63% and 95% above normal, WY1998 with 58% and 97% above normal, and WY2017 with 33% and 35% above normal over the two regions of interest mentioned above. These wet winters are punctuated by multi-year-long dry periods, giving rise to the so-called 'whiplash' events[36]. This is highlighted by the below-normal precipitation period between the WYs of 2012 to 2016 (~30-35% below normal in southern California, Fig. 3b) only to be relieved by the intense wet winter of 2016–2017, or by the multi-year drought from WYs 2019 to 2022 which was greatly alleviated by the extreme wetness of WY 2023[6]. The historical precipitation record demonstrates the prominent role of interannual fluctuations (blue vertical bars, Fig. 3) in addition to possible modulation on decadal-to-multidecadal timescales (green curve, Fig. 3). This raises the potential question: If the precipitation response shows a prominent role of high (interannual) as well as low frequency (decadal-to-multidecadal) variability, should these sources also be represented in the forcing function? This hypothesis serves as a guiding principle and a primary motivation for the construction of our seasonal precipitation prediction model.

### Precipitation hindcast skill assessment

The forecast skill of the MLMS−SST model over the western United States during the boreal winter (November to March) is shown in Fig. 4 applying *n*-fold cross-validation and displaying results over the recent five decades ranging from 1969 to 2018. Furthermore, forecast skill sensitivity is assessed for the number of SST predictor modes (*k*) under consideration. The first

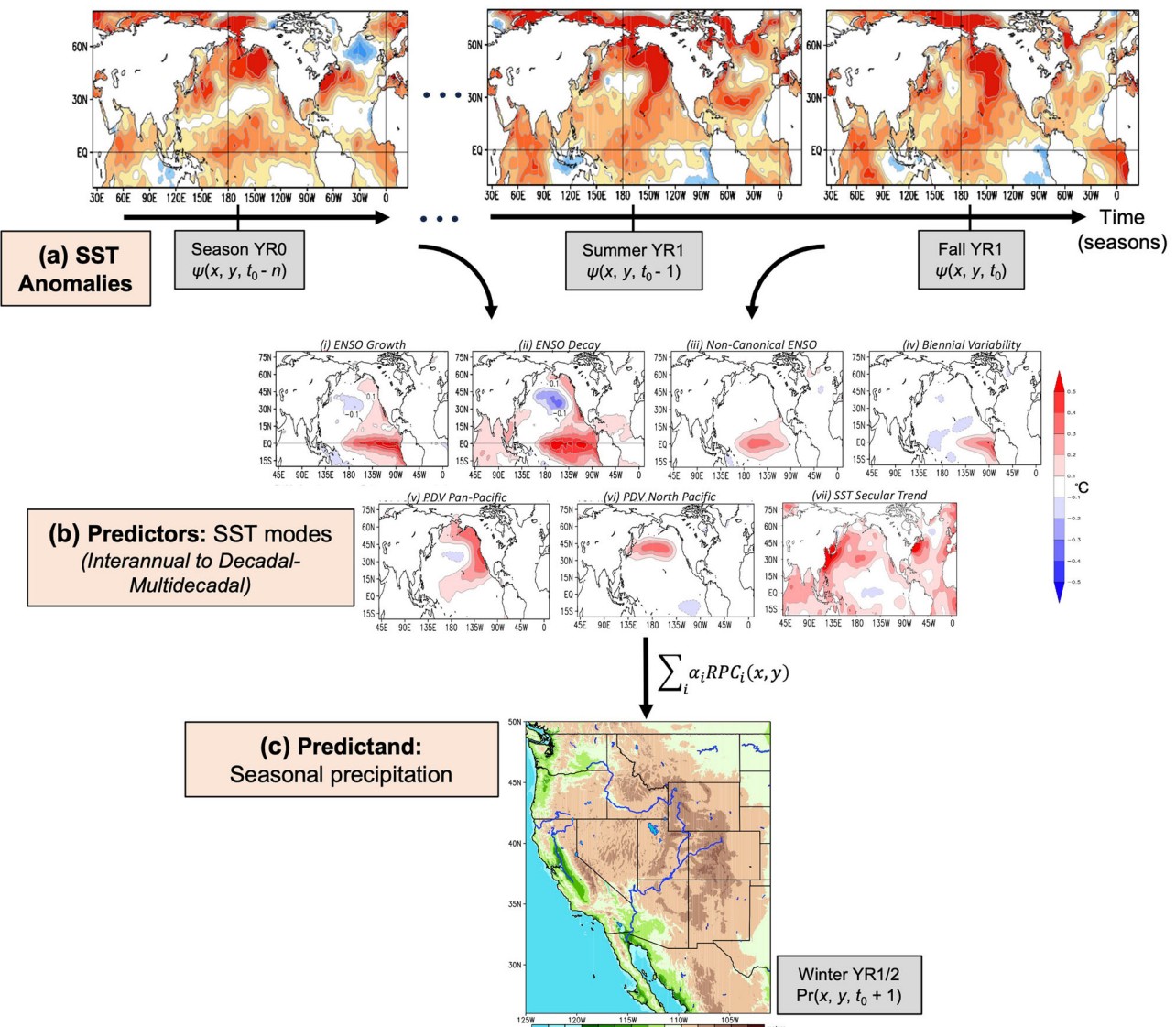

**Fig. 1 | Schematic of the seasonal precipitation forecast methodology based on evolving SST modes of variability.** Antecedent SST fields, $\psi(x, y)$, from multiple past seasons ($t_0$, $t_0 -1$, $t_0-2$, …, $t_0-n$), illustrated in (**a**), are projected onto predictor variables, shown in (**b**), extracted from an extended-EOF analysis of observed SST anomalies. The derived modes of variability comprise natural variability ranging from interannual to decadal-multidecadal timescales as well as the secular trend. A statistical model is trained by leveraging the lagged relationships between these predictor modes and the predictand, which is precipitation anomaly, Pr ($x$, $y$), over the western U.S. in the following winter season ($t_0 + 1$), shown in (**c**).

row considers four modes comprising the ENSO variability, i.e., $k = 4$, whereas the following rows consider the incremental value of adding the secular trend mode and sources of predictability dominant on decadal-multidecadal timescales in the Pacific ($k = 7$, second row) and in the Atlantic ($k = 11$, third row) respectively. Please note that four ENSO modes were required here in order to characterize the full extent (spatiotemporal progression from growth to peak and subsequently to its decay phase) and multiple flavors (canonical/East-Pacific vs noncanonical/Modoki) of ENSO variability (more details in the Methods section). The spatial extent of higher forecast skill generally expands with an increase in the number of predictor modes in the Pacific, i.e., from $k = 4$ to $k = 7$. This is evident also from the area-averaged correlations between forecasted and observed precipitation anomalies over the various hydrologic basins in the western U.S. (namely, northern and southern California, Pacific Northwest, Upper and Lower Colorado basins, and the Great Basin) reported in Tables 1–3. For instance, during the 1969–1978 period, the ENSO-only forecast ($k = 4$) scheme has an anomaly correlation of +0.4 over northern California and +0.48 over the Pacific Northwest (Table 1), which increases to +0.64 and +0.59,

respectively, with the $k = 7$ scheme (Table 2), that considers the influence of ENSO in addition to decadal variability in the Pacific Ocean. An examination of the forecast skill across the five decades under consideration (Fig. 4) reveals that the first and third decades, namely 1969−1978 and 1989−1998, were the most skillful ones closely followed by the period of 1999−2008, in terms of greatest coverage of area with anomaly correlation skill > +0.4, e.g., over the Pacific Northwest, California, and Lower Colorado basins. The additional consideration of decadal-multidecadal modes of variability in the Atlantic ($k = 11$) results in little to no improvement in skill (Table 3) beyond that obtained from the Pacific domain ($k = 7$). Our findings suggest that consideration of the combined influence of interannual modulations as well as decadal-multidecadal variability in the global oceans results in maximization of forecast skill and provides a promising new approach to improve seasonal prediction of winter precipitation across the western U.S.

### Rationale behind the selection of multiple temporal lags
To understand the impact of the number of temporal lags on seasonal prediction skill, sensitivity tests are conducted and shown in Fig. 5. Hindcast

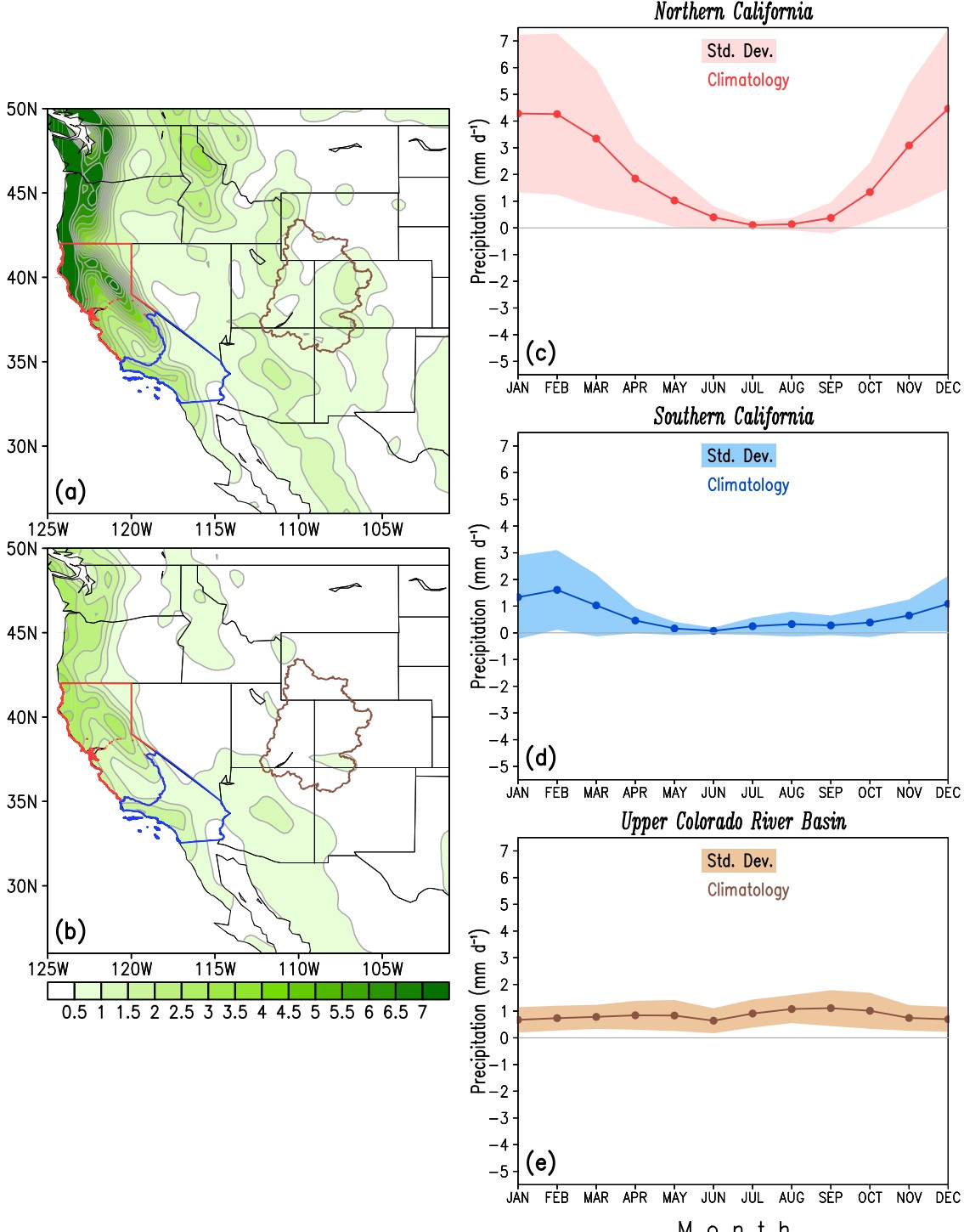

**Fig. 2 | Climatological winter precipitation and associated variability in the western U.S. a**, **b** The spatial distributions of seasonal winter mean precipitation (mm day$^{-1}$) and its standard deviation (mm day$^{-1}$) respectively, obtained from the GPCC version 2020 dataset. **c–e** The variability in observed precipitation for three constituent hydrologic basins—northern and southern California, and the Upper Colorado River basin, also outlined in (**a**) and (**b**). The solid line in each subpanel denotes the climatology from observations, while the shading represents their ±1 standard deviations. The period of analysis is 1981–2010.

skill is obtained from an *n*-fold cross-validation analysis and is presented here in terms of anomaly correlations (top row) and mean absolute error (MAE; bottom row) for the period of 1969−2018. Sensitivity is assessed for the number of seasonal lags considered in the MLMS−SST model. For example, a 1-season lag considers only the influence of SST predictors from a single antecedent season, whereas the 3- or 5-season lags consider the influence of the evolution of predictors over multiple antecedent seasons.

The hindcast maps reveal limited skill in forecasting winter precipitation when only a small number of antecedent lags are considered. When the number of seasonal lags is ≤3 (Fig. 5a, b), the areas with statistically significant correlation skill (~0.3) are limited to parts of the Desert Southwest and extend into Sonora (Mexico), whereas most of the western U.S. is devoid of any noticeable skill (i.e., anomaly correlations <0.2). With an increase in the number of

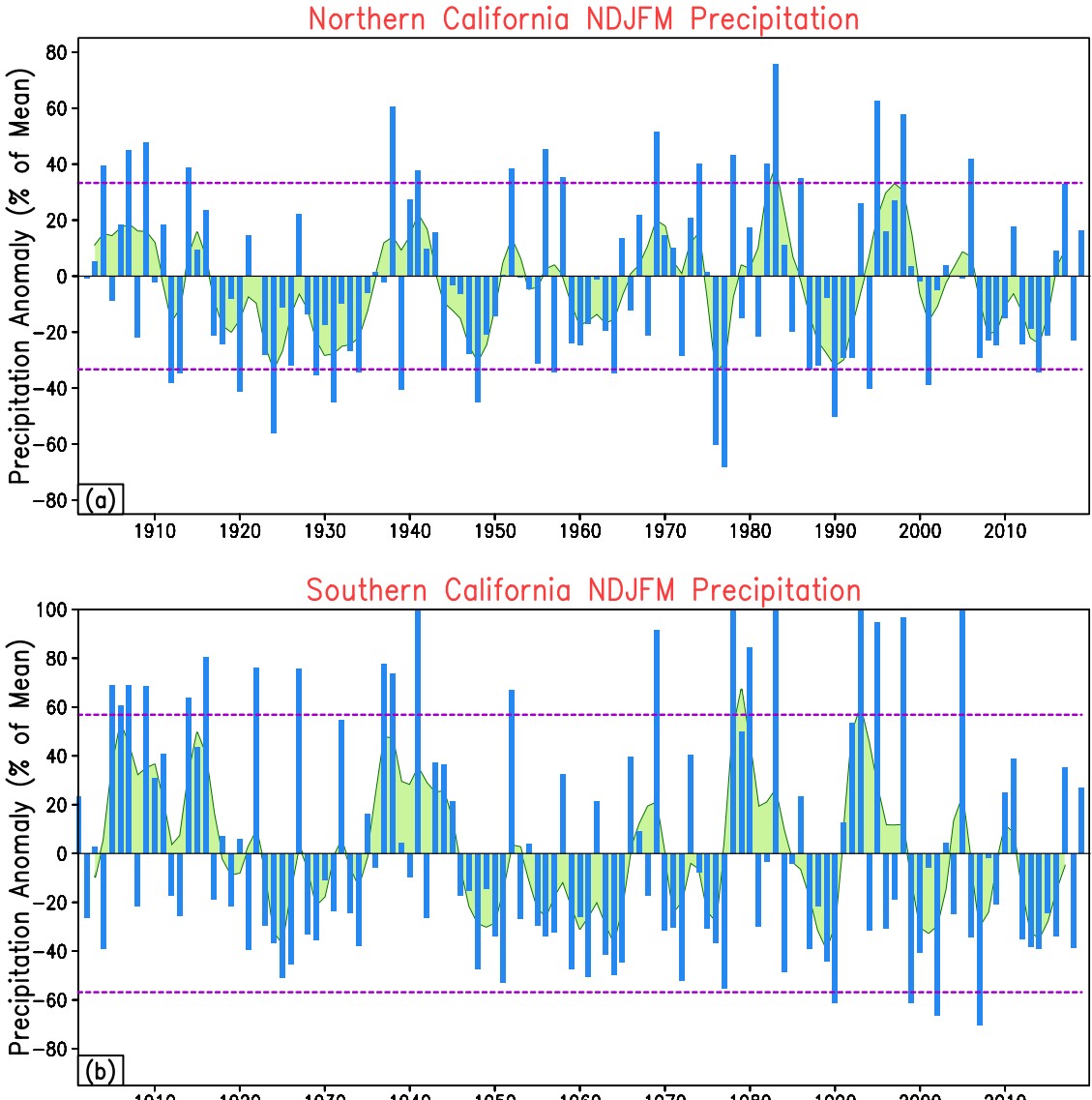

**Fig. 3 | Observed time series of winter precipitation in northern and southern California.** The precipitation anomaly time series for northern California (**a**) and southern California (**b**) are constructed and depicted (as blue bars) in terms of their deviation (in %) from its long-term mean. The time series of precipitation is also temporally smoothed using the LOESS filter and shown as the green shaded curve. The dashed violet lines represent the ±1 standard deviation in observed precipitation anomalies. Please refer to Fig. 2a and b (northern California marked in red, and southern California outlined in blue) for spatial extents of these two regions. The period of analysis is 1900–2018.

antecedent seasonal lags, regions of elevated forecast skill significantly expand across the western U.S. (Fig. 5c). Anomaly correlations of the order of +0.3−0.4 populate much of northern and southern California, Oregon, and Idaho, and extend into the Upper Colorado River basin. The MAE maps (Fig. 5d–f) exhibit a similar spatial structure and magnitude of bias across the number of analyzed temporal lags. The errors in the MLMS−SST model over the hindcast period are mostly concentrated over the coastal regions of the U.S. West Coast with typical MAE values ranging from 0.5 to 1.0 mm/day, and errors are smaller in the Intermountain West (0.25−0.5 mm/day). These results confirm our initial hypothesis that incorporating predictors from multiple previous seasons is crucial, as it leads to a significant improvement in forecast accuracy while maintaining similar bias structure and magnitude. Sensitivity tests conducted using 7-season lags and beyond showed a decrease in hindcast skill (Supplementary Fig. 2), which motivated our decision to employ the five-season temporal-lagged strategy in the present study.

## Skill comparison with dynamical and statistical forecast models

The outputs of the coupled ocean-atmosphere dynamical models from the North American Multi-Model Ensemble (NMME) project are analyzed to compare their predictive skills with the MLMS−SST model. The winter precipitation anomalies predicted by individual models are compared with observations over the common overlapping period (winters 1982–1983 to 2010–2011), and their hindcast performance is reported for six broad hydrologic basins of interest encompassing the western U.S. in Fig. 6. All four NMME dynamical models exhibit modest correlations in the range of +0.1−0.25 for four of the six HUC basins — northern California, the Pacific Northwest, the Upper Colorado River basin, and the Great Basin. Over the other two HUC basins, southern California and the Lower Colorado River basin, most NMME models also demonstrate weak predictive skill, however with the notable exception of the GFDL FLOR-B01 model, which has correlations of +0.52 and +0.6 respectively over these regions. A comparative assessment of the hindcast performance of the MLMS−SST model against the dynamical NMME models reveals that the MLMS−SST model consistently outperforms their multi-model ensemble mean in all six

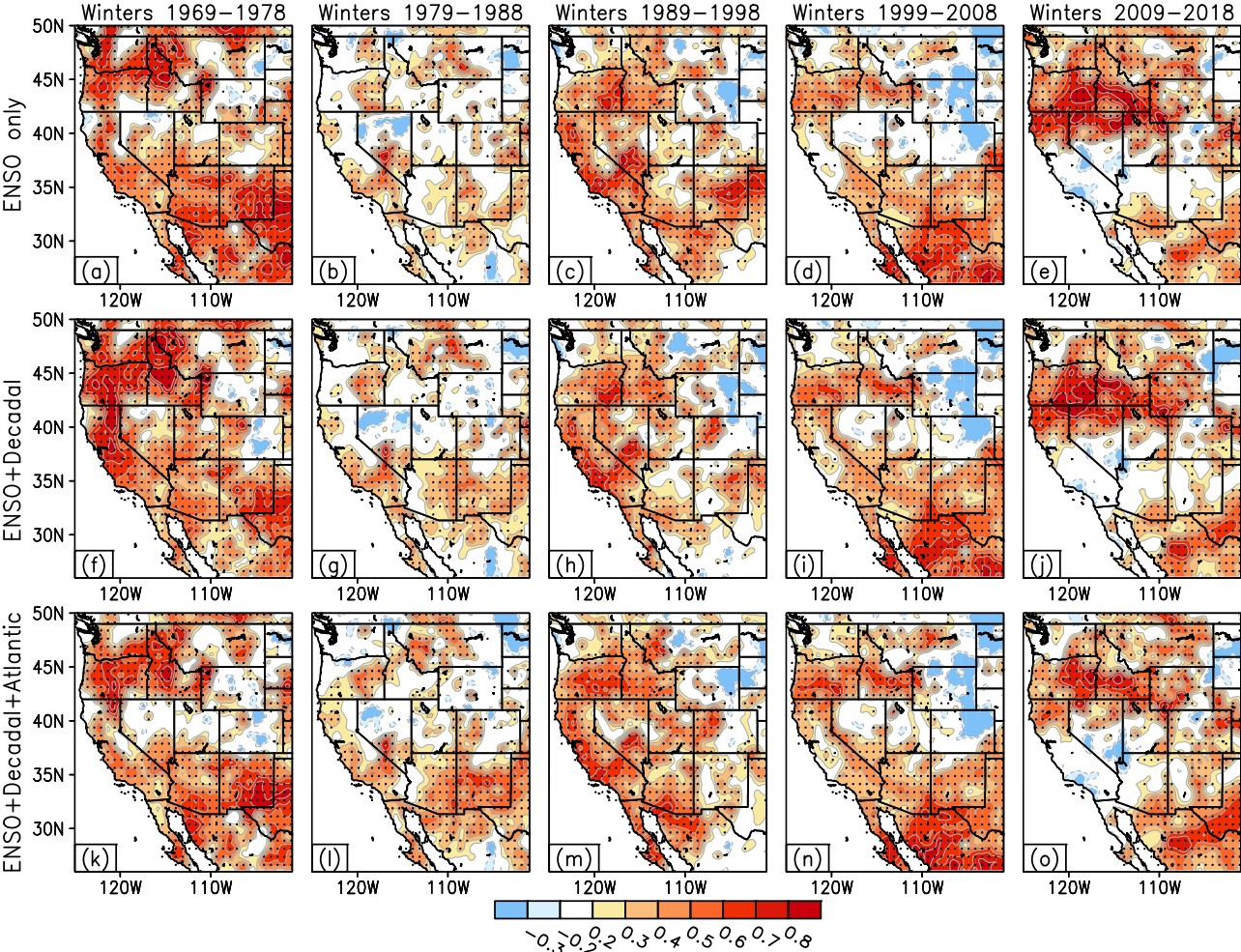

**Fig. 4 | Hindcast skill of the MLMS−SST model across the western U.S.** Cross-validation analyses over independent hindcast periods yield correlations between the observed and model-predicted winter precipitation anomalies. **a−e** Skill scores obtained when only ENSO modes of variability are considered in the predictor set, whereas subsequent panels depict skill with additional contributions from the secular trend and decadal variability modes in the Pacific (**f−j**) and in the Atlantic (**k−o**). Correlations that are statistically significant at the 95% confidence level are stippled in black. The correlation values are contoured and shaded in red at intervals of 0.1 when ≥ +0.2, and in blue when ≤ −0.2.

western U.S. regions. Specifically, the MLMS−SST model displays the highest hindcast correlation skill over northern California and the Pacific Northwest compared to all other models. Additionally, it closely matches the best-performing model, GFDL FLOR-B01, in the Upper Colorado River basin. Spatial maps illustrating the hindcast skill (Supplementary Fig. 3) show the extent of the MLMS−SST model's coverage of areas with robust correlation skill (anomaly correlations > +0.3) over California, Oregon, Idaho, and most of the Upper and Lower Colorado River basins.

The MLMS−SST model is further evaluated against other statistical model benchmarks. In the previous section, the model's hindcast skill was analyzed with longer temporal lags (e.g., 5-season lag), revealing a significant outperformance over statistical approaches that rely solely on predictors from the previous antecedent season (i.e., a 1-season lag), which is the methodology employed in the CCA statistical model discussed earlier[11]. Our results (Fig. 5) indicate a significant improvement in forecast skill when utilizing a longer temporally lagged window for predictor variables, as opposed to using predictive information from a single snapshot of time, as done in the CCA method. Additionally, the hindcast skill of the MLMS−SST model is compared against another statistical model benchmark, the persistence forecast, over the period of 1969−2018 (Supplementary Fig. 4). Our proposed model outperforms the benchmark by as much as 60% in terms of forecast accuracy (see Methods section for related computation of MSE-based skill scores) over broad swaths of the western U.S., highlighting

its potential to provide more skillful forecasts of seasonal winter precipitation.

## Dissecting individual seasonal forecasts & understanding the modal attribution process

The previous sections have highlighted that the trained MLMS−SST model exhibits improvements relative to existing methods in predicting seasonal precipitation across key regions experiencing winter precipitation. In this section, we delve into individual seasonal forecasts to elucidate the model's modal attribution process, which favors specific outcomes at these longer lead times, such as drier or wetter than normal conditions in a particular winter. In Fig. 7, we present a case study from the winter of 2021–2022 that was associated with intense dry conditions in the region, with California experiencing its driest January through April period on record since 1895[37]. In fact, the state entered this winter season with severely dry antecedent conditions following its second driest water year (October 1, 2020, through September 30, 2021), which necessitated very low allocations for the State Water Project and other large water projects. The MLMS−SST model's forecast (Fig. 7b), issued in October 2021, using antecedent predictor information, efficiently predicted the observed drier-than-normal precipitation conditions from November to March 2021–2022 (Fig. 7c). The observed SST anomalies in the five antecedent seasons, which inform the MLMS−SST model's forecast, are illustrated in Supplementary Fig. 5.

**Table 1 | Correlations between observed and predicted precipitation anomalies by the MLMS–SST model using only four predictor modes constituting ENSO variability (k = 4)**

|  | Period: 1969-78 | Period: 1979-88 | Period: 1989-98 | Period: 1999-08 | Period: 2009-18 |
|---|---|---|---|---|---|
| Northern California | **0.40** | 0.17 | **0.50** | 0.21 | **0.31** |
| Southern California | **0.42** | **0.34** | **0.54** | **0.38** | -0.11 |
| Pacific Northwest | **0.48** | 0.12 | **0.34** | **0.27** | **0.49** |
| Upper Colorado | **0.33** | 0.15 | 0.25 | 0.09 | **0.33** |
| Lower Colorado | **0.55** | 0.21 | **0.39** | **0.38** | 0.14 |
| Great Basin | **0.27** | 0.09 | **0.35** | 0.08 | **0.41** |

Results are computed from *n*-fold cross-validation analysis followed by area-averaging across various constituent HUC basins in the western U.S. over the past five decades. Correlations statistically significant at the 95% confidence level are highlighted in **bold**. Additionally, the highest correlations for each region and period comparing across Tables 1–3 are highlighted in red.

**Table 2 | Skill improvement by including additional Pacific decadal variability modes (k = 7)**

|  | Period: 1969-78 | Period: 1979-88 | Period: 1989-98 | Period: 1999-08 | Period: 2009-18 |
|---|---|---|---|---|---|
| Northern California | **0.64** | 0.22 | **0.52** | **0.31** | **0.33** |
| Southern California | **0.44** | **0.38** | **0.51** | **0.39** | -0.09 |
| Pacific Northwest | **0.59** | 0.14 | **0.33** | **0.29** | **0.46** |
| Upper Colorado | **0.41** | 0.18 | **0.28** | 0.17 | **0.33** |
| Lower Colorado | **0.46** | **0.29** | **0.31** | **0.40** | 0.14 |
| Great Basin | **0.39** | 0.03 | **0.38** | 0.20 | **0.36** |

As in Table 1, but here correlations are computed for a forecasting strategythat additionally incorporates the two Pacific decadal variability modes as well as the secular trend.

**Table 3 | Little-to-no skill improvement when combining Atlantic SST modes (k = 11)**

|  | Period: 1969-78 | Period: 1979-88 | Period: 1989-98 | Period: 1999-08 | Period: 2009-18 |
|---|---|---|---|---|---|
| Northern California | **0.37** | 0.25 | **0.52** | 0.33 | 0.23 |
| Southern California | **0.39** | **0.34** | **0.51** | **0.38** | -0.12 |
| Pacific Northwest | **0.47** | 0.12 | **0.38** | 0.35 | **0.39** |
| Upper Colorado | 0.24 | 0.22 | **0.38** | 0.17 | **0.30** |
| Lower Colorado | **0.50** | 0.35 | **0.46** | **0.40** | 0.16 |
| Great Basin | 0.18 | 0.09 | **0.36** | 0.19 | 0.22 |

As in Table 1, but here the forecasting strategy considers four additional modes of SST variability from the Atlantic. Refer to Supplementary Fig. 1 for the characteristic spatial structure of SST anomalies associated with the mature phase of these modes.

Negative values indicative of cold SST anomalies associated with a La Niña event populate much of the central and eastern tropical Pacific Ocean during fall of 2020 through the spring of 2021 and again during the fall of 2021. If the forecast is based solely on the interannual modes of ENSO variability, we obtain a typical La Niña precipitation response (Fig. 7a), characterized by a wet-north and dry-south pattern, at odds with actual observations. However, in addition to the ongoing La Niña episode, the multi-season lagged SST predictor field (Supplementary Fig. 5) reveals another interesting concurrent phenomenon: persistent warm SSTs were present in the North Pacific Ocean, bearing a striking resemblance to the Pacific decadal variability (PDV) modes, particularly the PDV North Pacific mode (Supplementary Fig. 1; middle panel). Additional consideration of the influence of these lower frequency modes which vary on decadal-to-multidecadal timescales in the Pacific, a key aspect of the MLMS–SST model, yields a seasonal forecast (Fig. 7b) that skillfully captures the observed drier-than-normal conditions, with dry maxima focused over northern California and coastal Oregon. In summary, a canonical La Niña development unfolded over the preceding seasons leading up to the winter of 2021–2022 upon background decadal variability signals in the Pacific. The incorporation of the spatiotemporal evolution of these diverse SST modes of variability affords a more skillful reconstruction of their resulting influence on precipitation over the western U.S.

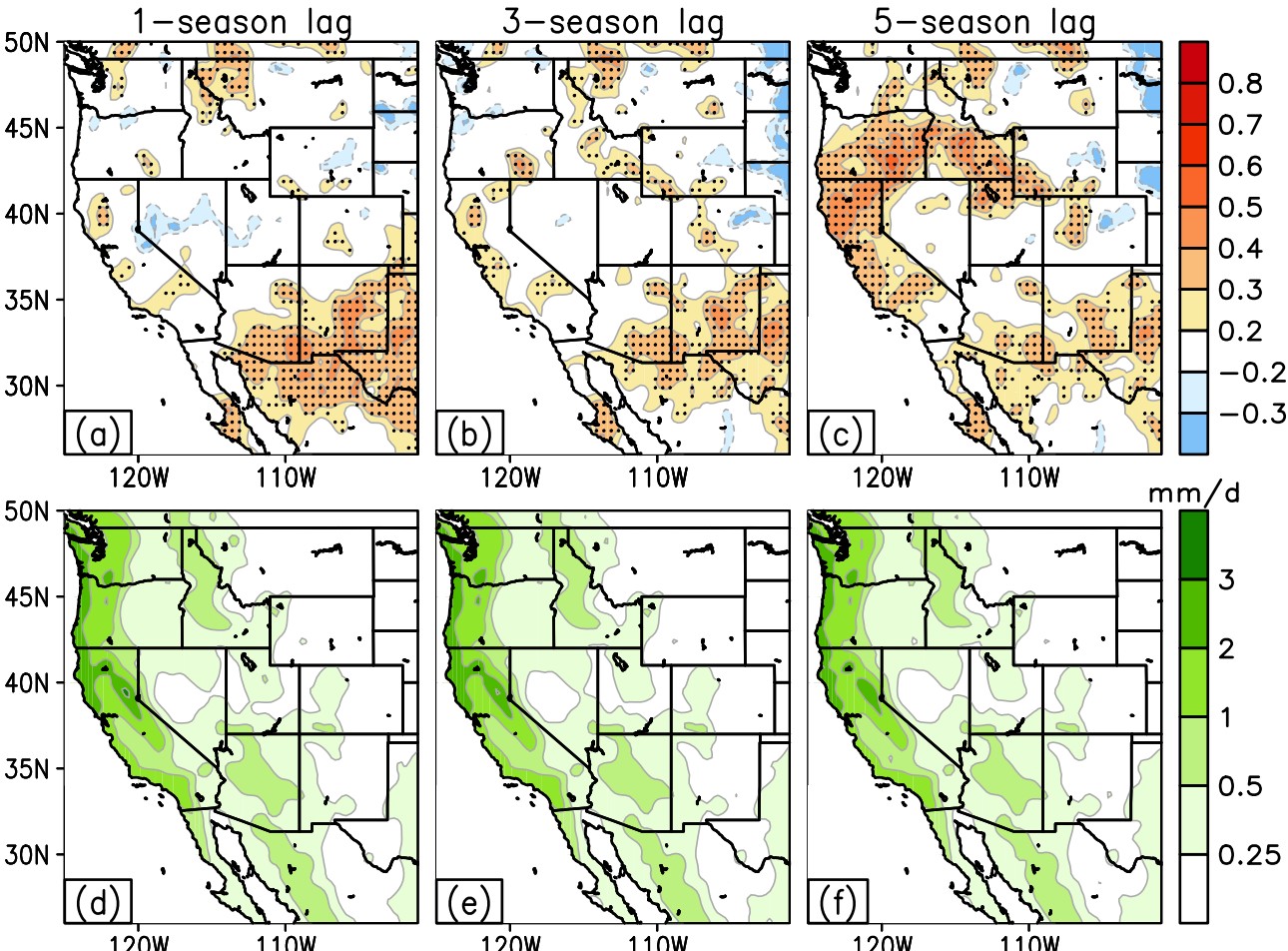

**Fig. 5 | Sensitivity tests highlighting the efficacy of antecedent, multi-season lagged predictors.** The hindcast skill scores are obtained from *n*-fold cross-validation analyses and displayed as a function of the number of temporal lags employed in the MLMS–SST model, namely 1-, 3-, or, 5-seasons preceding the time of forecast issuance. Skill is depicted here in terms of anomaly correlations (**a**–**c**) as well as mean absolute error (MAE in mm day$^{-1}$; **d**–**f** between the model-predicted and observed winter precipitation anomalies. The hindcast period of assessment is 1969–2018. The correlations that are statistically significant at the 95% confidence level are stippled in black in (**a**)–(**c**).

Next, we present seasonal precipitation forecasts from the MLMS-SST model and related verification in Fig. 8 for the most recent winters of 2017–2018 through 2023–2024. To generate the model forecast during these winter seasons, the model leverages the observed SST anomalies during the five antecedent seasons prior to winter in order to generate the projections of the predictor SST modes. From Fig. 8, we notice that the MLMS-SST model skillfully captures the observed precipitation conditions in at least five of the seven most recent winter seasons over the western U.S., with the notable exception of winter 2022–2023 which was a challenging water year for most seasonal forecast systems[6]. The model forecast (first and third rows of Fig. 8) is characterized by accurate depictions of both below-normal (e.g., winter 2017–2018, or 2020–2021) and above-normal (e.g., winter 2023–2024) precipitation conditions in the western U.S.

**Potential physical mechanisms governing the interaction between multi-season lagged SSTs and precipitation changes**

While the focus of the study is on improving the prospects for seasonal precipitation forecasting, the possible mechanisms by which these antecedent, multi-season lagged SST anomalies in the Pacific influence precipitation changes in the western U.S. are briefly investigated in this section. These SST influence mechanisms are examined in Fig. 9 via changes in the streamfunction field at 850-hPa ($\Psi_{850}$), which characterizes the lower-tropospheric rotational circulation response, and resulting precipitation during winter. The anomalies are obtained from their temporally lagged

regressions on the antecedent SST principal components, which are obtained by sampling over multiple past seasons' SST anomalies (five past seasons in this case).

The top panel illustrates the ENSO influence mechanisms via the consideration of contributions from the canonical modes of ENSO variability (ENSO Growth and Decay) in Fig. 9a with the inclusion of contributions from the Non-canonical ENSO and ENSO-Biennial modes in Fig. 9b. The $\Psi_{850}$ anomalies succinctly capture how these temporally lagged interannual modes of SST variability influence regional precipitation: an intense cyclonic circulation (with $\Psi_{850}$ anomalies of the order of $-1.8 \times 10^6$ to $-2.1 \times 10^6$ m$^2$ s$^{-1}$ unit SST PC$^{-1}$) situated just off the west coast of North America facilitates onshore moisture transport leading to orography-mediated precipitation surplus (green shading) in the western U.S. In contrast, the northern coastal regions of British Columbia and Alaska experience below-normal precipitation (brown shading), consistent with the offshore flow associated with the above-mentioned circulation anomaly pattern. The El Niño-related response is also characterized by a pair of cyclones straddling the equator in the central Pacific (where $\Psi_{850}$ anomalies of the order of $0.3 \times 10^6$ to $0.6 \times 10^6$ m$^2$ s$^{-1}$ unit SST PC$^{-1}$), resulting in enhanced equatorial westerlies. Similarly, the modes of Pacific decadal variability, namely Pan-Pacific (Fig. 9c) and North Pacific (Fig. 9d), are found to modulate the lower tropospheric flow, resulting in an anomalous cyclonic circulation over the Northeast Pacific, albeit with weaker magnitude ($\Psi_{850}$ anomalies of the order of $-0.6 \times 10^6$ to $-0.9 \times 10^6$ m$^2$ s$^{-1}$ unit SST PC$^{-1}$) compared to the ENSO

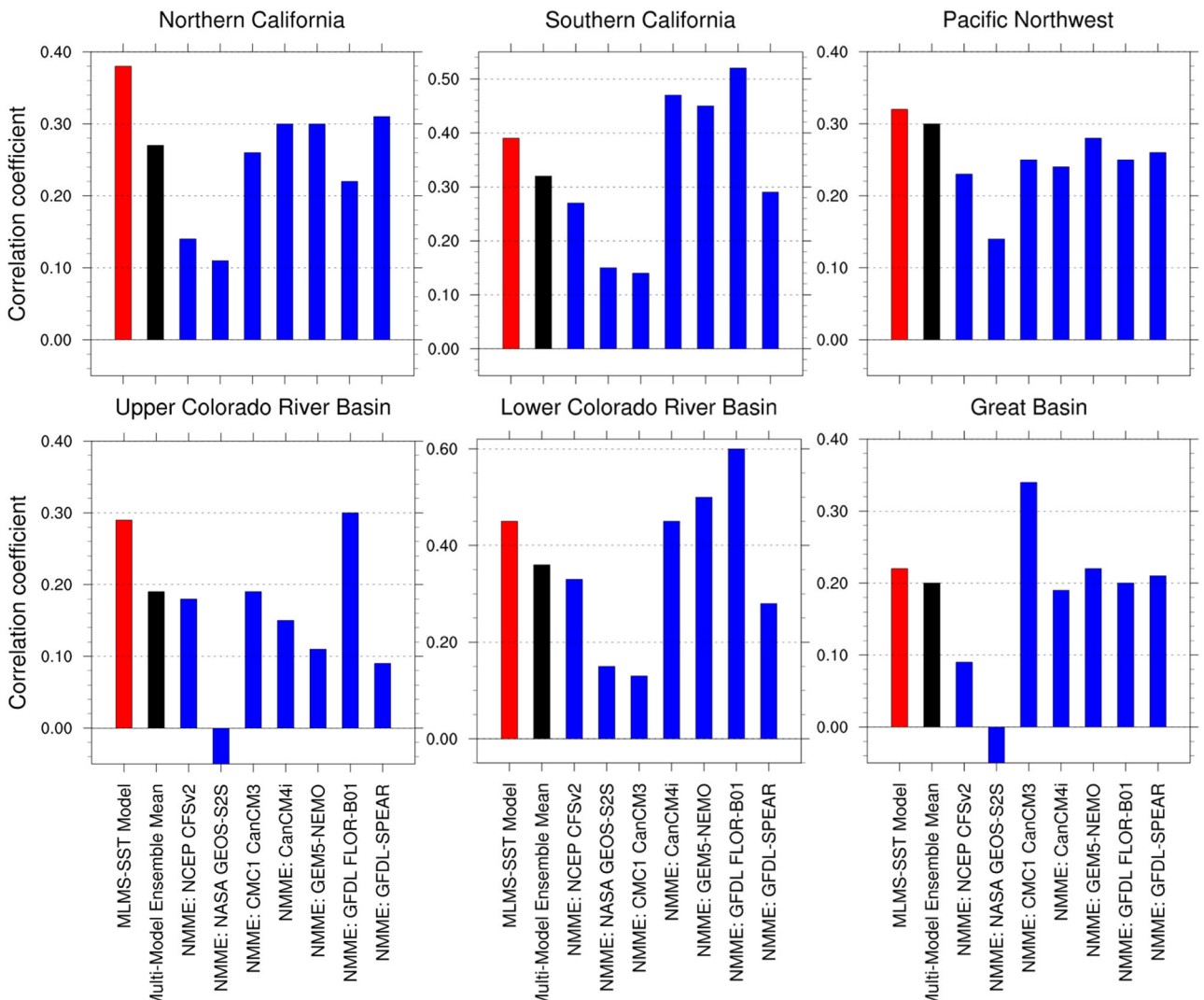

**Fig. 6 | Hindcast skill of MLMS-SST model compared with the NMME dynamical models and their multi-model ensemble mean over constituent HUC basins in the western U.S.** The dynamical models assessed here include the NCEP-CFSv2, NASA GEOS-S2S, CMC CanCM3, CanCM4i, GEM5-NEMO, GFDL FLOR-B01 and GFDL-SPEAR, with hindcasts initialized in October for the November through March winter season. These dynamical models are shown in blue with their multi-model ensemble in black, and the proposed MLMS-SST model in red. The comparative assessment is performed for the common overlapping period of available hindcasts —winters 1982-83 through 2010–11 (with the exception of GFDL-SPEAR, available only from winter 1991-92). The correlations reported in each case are the area-averaged values computed over continental grid points.

influence. The related onshore moisture transport and interaction with regional orography leads to the observed positive precipitation anomalies in the western U.S. These mechanistic findings are in agreement with previous studies[38,39], which have identified orographic processes as the primary driver of precipitation in the region with onshore moisture fluxes driven by synoptic-scale circulation anomalies.

## Discussion

Lead time is of paramount importance for efficient management of water resources and skillful forecasts at the longer lead times stand to benefit some of the most impactful and expensive operational decisions[40,41]. Much of the western U.S., specifically California and the Colorado River basin, has been facing severe and persistent drought for much of the 21st century, with a few wet winters like those in 2016–2017 and 2022–2023 providing brief hydrologic relief. These extreme conditions make it exceptionally challenging to manage resources and mitigate the impacts of climatic extremes. Reliable longer-lead predictions of seasonal precipitation have the potential to provide much-needed assistance in managing drought and flood risks, motivating our current study.

Our research focuses on developing and applying the MLMS−SST model to predict seasonal winter precipitation in the western U.S. The model leverages the oceanic memory of SST predictors from several antecedent seasonal leads, giving the 'Multi-Lead' or 'ML' part of the MLMS acronym. We find that the predictive skill of the developed model increases with an increase in the number of temporal lags considered before the time of forecast issuance (Fig. 5). So, the question arises: why are SST predictors from the most recent season not effective predictors of seasonal precipitation, and why is it necessary to employ multiple temporal lags to capture predictive information? This can be explained by a hypothetical case study (Supplementary Fig. 6): say there are two modes of SST variability in the Pacific, one high-frequency or faster-evolving mode, and another low-frequency or slowly evolving mode. Both modes exhibit a similar spatial pattern of SST anomalies, characterized by a north-south dipole featuring colder anomalies in the North Pacific and warmer anomalies in the equatorial Pacific. Now, if at any instant in time ($t_0 = T$) the observed SST anomaly field resembles this dipole, it becomes critical for seasonal prediction purposes (when predicting precipitation at $t_1 = T + n$ seasons) to accurately pinpoint which of the above two modes

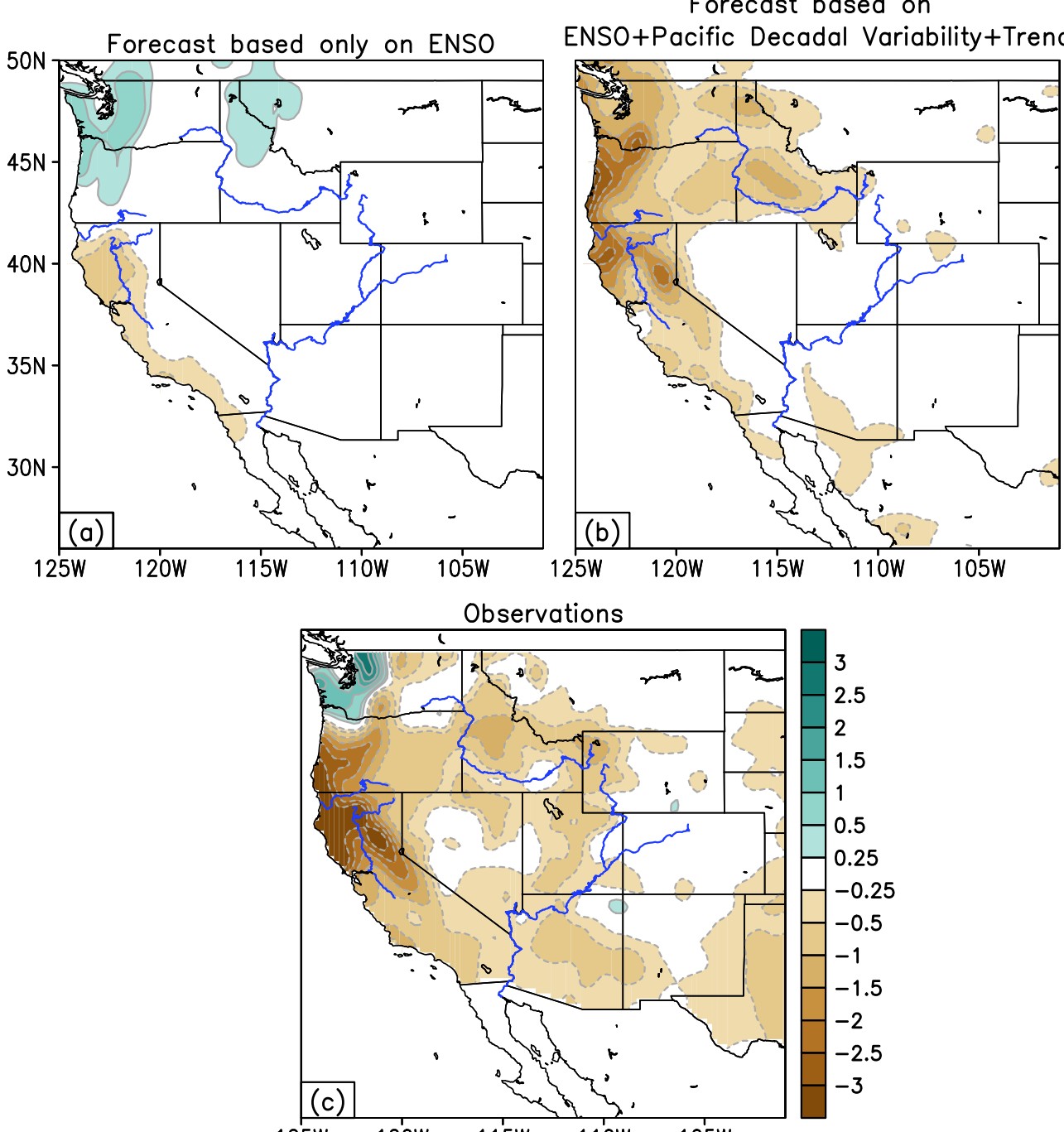

**Fig. 7 | Explaining individual seasonal precipitation forecasts through modal attribution.** A case study is presented for the winter of 2021–2022. **a, b** Two versions of the MLMS–SST model forecast, the former considers contribution coming only from ENSO, whereas the latter additionally incorporates sources of decadal-to- multidecadal variability in the Pacific. This winter season was associated with severe drier-than-normal conditions in the western U.S., as illustrated in (**c**). Green and brown shading denotes wetter-than-normal and drier-than-normal precipitation conditions respectively. Major rivers in the western U.S. are outlined in blue.

is active. If the higher frequency mode is active, it might change phase within $n$ seasons, whereas if the lower frequency mode is dominant, it will maintain its structure. Each of these scenarios will be associated with distinct regional precipitation impacts. Therefore, it is imperative to focus on the past multi-season structure of the predictor modes rather than just the preceding season for an accurate attribution of the dominant mode. Our study advances upon related works[11,12,42]. Instead of just exploiting the lagged predictor-predictand relationship at various discrete snap- shots of time (e.g., 1, 6, 12, or 18 months prior), as done in these prior studies, our framework tracks the entire progression of SST predictor

modes in both space and time over multiple past seasons, resulting in maximization of forecast performance.

Additionally, this study incorporates multiple sources of predictability encompassing both natural variability and the secular trend in the global oceans, providing the 'Multi-Source' or 'MS' part of the MLMS acronym. These predictor modes, identified from an extended-EOF analysis of the 20th–21st century global SST variability, exhibit dominant time periods ranging from interannual to decadal-to-multidecadal timescales. Recent studies[28,29] have revealed that only about 25% of the variability in California winter precipitation can be explained by ENSO. This finding supports our

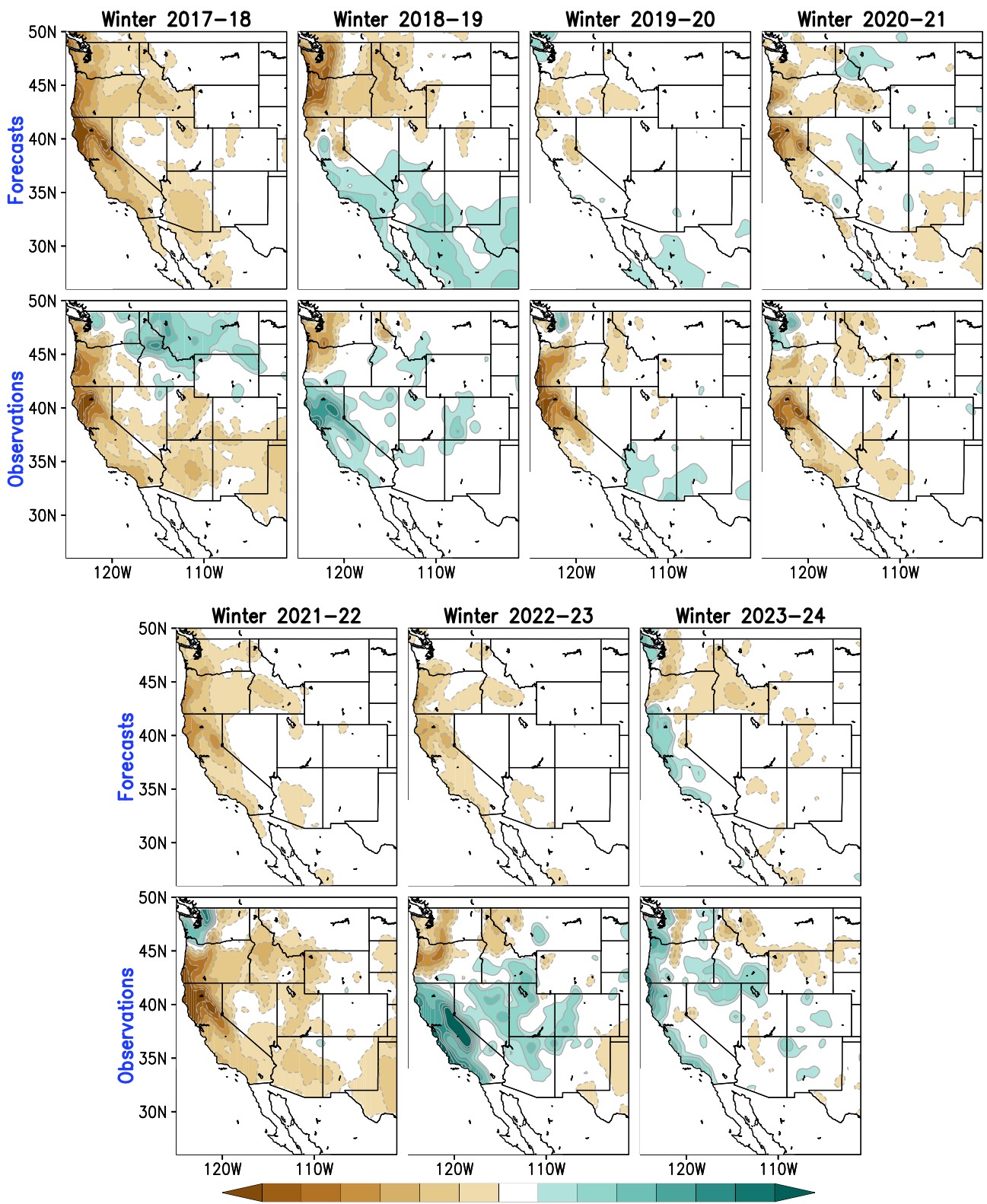

**Fig. 8 | Seasonal precipitation forecasts from the MLMS-SST model and related verification for the period of winters 2017−2018 through 2023−2024.** The observational verification is based on the NOAA CPC Unified dataset. Green and brown shading denotes positive (wetter-than-normal) and negative (drier-than-normal) precipitation anomalies respectively, in units of mm/day.

choice to include SST predictors that encompass not only interannual timescales but also longer timescales to account for sources of Pacific decadal variability. Sensitivity tests reveal that an increase in the number of predictor SST modes enhances the seasonal prediction skill (as

demonstrated in Fig. 4 and Tables 1 and 2) beyond that realized just based on ENSO.

Hindcast skill results over the period of assessment show that the MLMS−SST model can compete with or outperform leading dynamical

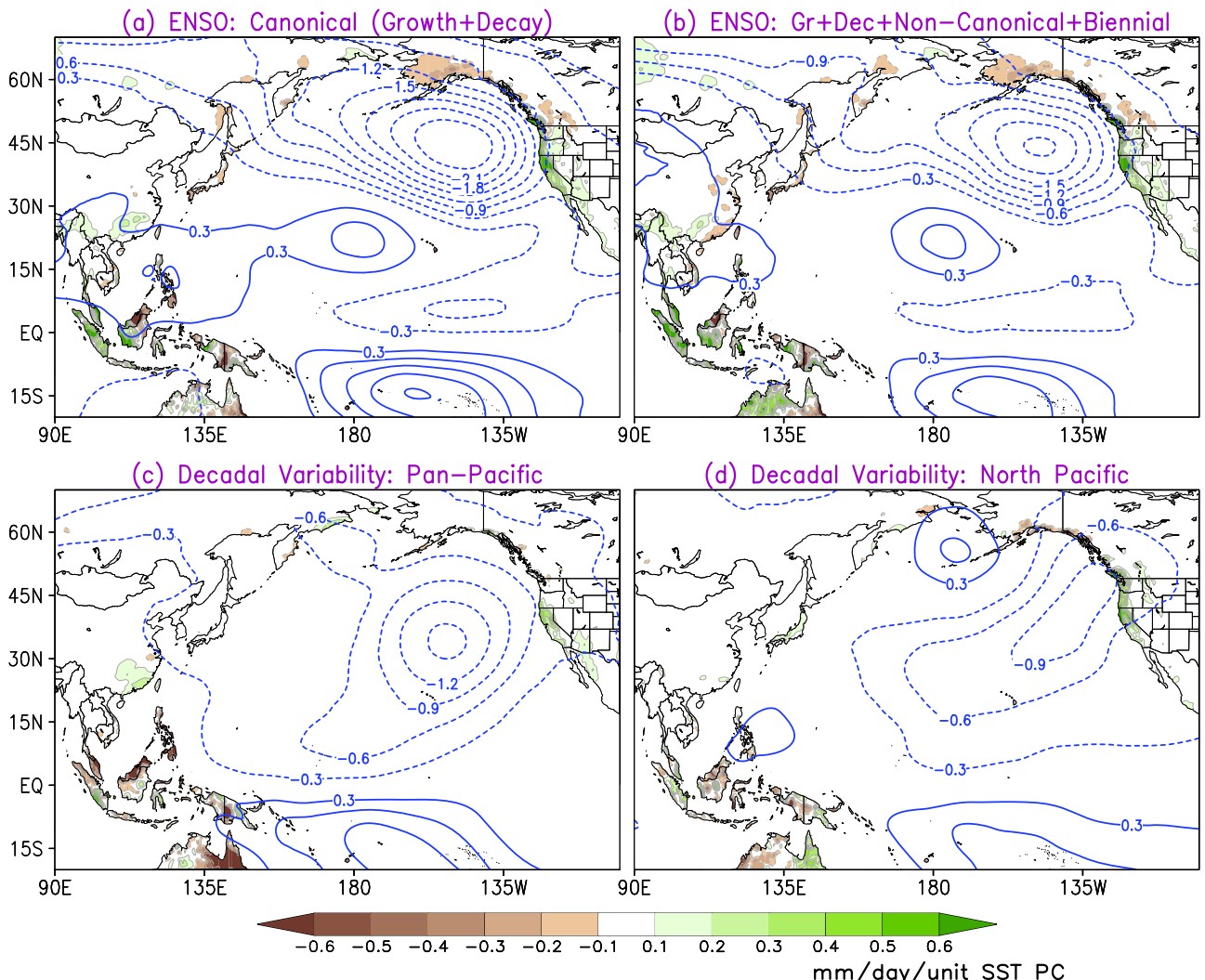

**Fig. 9 | SST-driven mechanisms influencing regional precipitation changes.** This influence is illustrated via anomalies in the 850-hPa streamfunction (contours) and resulting precipitation (shading) during the winter season. The anomalies are obtained from temporally lagged regressions of precipitation (from GPCC version 2020) and streamfunction (from ERA5) on the antecedent principal components (PCs) of the SST modes, extracted using SST anomalies over five preceding seasons. **a, b** The influence mechanisms of different combinations of the four constituent ENSO modes of variability, whereas subsequent panels depict those from the two decadal modes of variability in the Pacific (**c, d**). The period of analysis is 1948–2018. Green shading represents precipitation surplus, and brown shading indicates deficit; contour interval is 0.1 mm day$^{-1}$ per unit SST PC. Meanwhile, solid blue contours represent positive 850-hPa streamfunction anomalies and dashed blue contours denote negative 850-hPa streamfunction anomalies; the contour interval is $0.3 \times 10^6$ m$^2$ s$^{-1}$ per unit SST PC.

models as well as other statistical approaches, as measured by anomaly correlations and spatial coverage of areas with high skill. Notably, we find significant improvement in seasonal prediction skills using the MLMS−SST model in Northern California, Pacific Northwest, and the Upper Colorado River basin (Fig. 6 and Supplementary Fig. 3). The model affords the opportunity to enhance our understanding of individual forecasts by identifying the sources of variability that contribute to the precipitation forecast (Fig. 7). In contrast, conducting computationally expensive coupled modeling experiments would be necessary in dynamical models to quantify the specific influence of different boundary conditions on a forecast at long lead times, making the same task much more challenging. Additional precipitation hindcast skill assessments are conducted to ensure the model does not exhibit elevated skill due to sampling variability. These analyses utilizing predictors extracted from one-half of the analysis period for model training and testing on the other half (and vice versa) reveal no notable differences in realized skill (Supplementary Fig. 8).

Future research could explore additional avenues to further enhance seasonal precipitation forecasting skills. One such approach might entail incorporating additional sources of predictability in the Earth system (e.g.,

soil moisture, terrestrial snow cover, etc.) with similar long-term memory that could be leveraged for seasonal forecasts. However, efforts to accommodate multiple sources of predictive influences and their related interactions often result in overfitting[43]. This uncertainty can be tackled by using methods like the partial-least-square regression (PLSR), designed specifically to avoid the overfitting problem. Another option is to explore the application of machine learning algorithms, which also account for nonlinearity in process interactions. However, a challenge lies with the limited size of reliable observational and reanalysis data available for training such predictive models. To overcome this limitation, hybrid strategies could be employed, such as training models in the dynamical/climate model space[24], but then applying transfer learning[44] to update model weights using real-world observations.

## Summary

The work presents a method to leverage diverse sources of SST variability evolving over multiple past seasons to potentially improve the skill and utility of seasonal forecasting of western U.S. precipitation. The key findings of the study can be summarized as follows:

- Winter precipitation exhibits unique variability in the hydrologic basins of the western United States, especially over California (Fig. 2). In this region, the observed precipitation record over the past century shows prominent fluctuations on interannual as well as decadal-to-multidecadal timescales (Fig. 3), motivating our choice of inclusion of predictors varying across multiple frequencies in the developed prediction model.
- The seasonal precipitation forecast skill increases with an increase in the number of predictor SST modes. Specifically, the incorporation of Pacific decadal variability alongside interannual modulations of ENSO leads to enhanced forecast performance (Fig. 4, middle panel, and Fig. 7).
- Seasonal forecast skill improves with the inclusion of more temporal lags in the predictor set and is maximized with a 5-season temporal window preceding the time of forecast issuance (Fig. 5). This finding validates our initial hypothesis concerning the importance of adequately capturing the spatiotemporal evolution of predictor variables from multiple past seasons, rather than relying solely on the most recent season's data.
- The MLMS−SST model proposed in this study demonstrates skill in terms of spatial coverage, with broad swaths of the study domain exhibiting anomaly correlations > +0.4 during the independent hindcast assessment period (Fig. 4). The model hindcasts are skillful relative to dynamical and statistical baselines and are competitive with or superior to the NMME dynamical models, particularly in Northern California, Pacific Northwest, and the Upper Colorado River basin (Fig. 6).
- We attempt to uncover the potential mechanisms through which the SST modes of variability influence regional precipitation changes. The positive precipitation anomalies are shown to be generated by an intense cyclonic flow centered off the west coast of North America, which promotes onshore moisture transport and resulting orography-mediated precipitation in the western U.S. (Fig. 9).

Our study offers a promising new approach for the potential improvement of seasonal precipitation forecasting, which has been defined as a top priority in the Weather Research and Forecasting Innovation Act[45] passed by the 115th U.S. Congress as well as recognized internationally as a research priority by the World Meteorological Organization. The skill of existing seasonal forecasting systems is currently limited for winter precipitation in California and the Colorado River Basin, an important source of imported water supplies. To address this capability gap, our proposed methodology develops a forecasting tool that has the potential to aid water and agriculture managers, as well as emergency response planners, in their resource positioning and policy decisions[40], particularly those undertaken at the onset of the winter season in the water-stressed western U.S. Furthermore, it establishes an updated empirical benchmark that can be leveraged for future seasonal model development and evaluation studies.

## Methods
### Observational and reanalysis datasets
The analysis uses the U.K. Met Office's Hadley Centre Sea Ice and Sea Surface Temperature dataset (HadISST 1.1)[46], available globally at a monthly 1° resolution from 1900 to the present. Seasonal SSTs follow the boreal season definitions and anomalies are constructed by removing the long-term climatology of each season. The Global Precipitation Climatology Centre (GPCC)[47] monthly precipitation dataset, version 2020, available at a 0.25° by 0.25° spatial resolution is used in the study. It is a quality-controlled, land-surface precipitation dataset for the January 1891−December 2019 period featuring an augmented database of stations (derived from ~84,800 stations worldwide) and an enhanced analysis method (climatological infilling in data-sparse areas) among recent improvements. The observed precipitation is used in the construction of anomalies over the extended winter season (from November through the following March) and for lagged regressions on the identified SST predictor variables. The NOAA Climate Prediction Center (CPC)'s Unified CONUS dataset (CPC Unified)[48], available at 0.25° by 0.25° resolution over the domain 20.125°−49.875°N, 230.125°−304.875°E for the period January 1948 to the present, is also utilized in this work. Sensitivity tests related to the choice of the observational dataset revealed minimal to no differences in the realized precipitation hindcast skill (Supplementary Fig. 7).

We used the European Center for Medium-Range Weather Forecasts (ECMWF)'s ERA5 Reanalysis[49], which is a state-of-the-art global atmospheric reanalysis product, for the computation of the streamfunction ($\psi$). It is related to the $u$ and $v$ components of wind as follows:

$$u = -\frac{\partial \psi}{\partial y}, \text{ and } v = \frac{\partial \psi}{\partial x} \qquad (1)$$

The streamfunction fields at 850-hPa are analyzed to elucidate the physical mechanisms governing the SST influence on regional precipitation.

### Precipitation hindcasts from NMME dynamical model database
Dynamical modeling resources available from NOAA's NMME Project Phases I and II are analyzed in this study. As part of the NMME project, coupled atmosphere-ocean models from leading modeling centers provide retrospective as well as real-time forecasts at a common spatial resolution of 1° by 1°. Supplementary Table 1 provides details on the NMME models analyzed, the number of ensemble members, and available lead times. These models are initialized monthly and run out to long lead times (9–12 months into the future). The database includes retrospective hindcasts spanning ~30 years for most models. In this study, all available model hindcasts initialized in October for November through March winter precipitation were utilized, following similar evaluation setups used in previous studies[24,28].

### Spatiotemporal SST analysis enabling prediction
This section provides an overview of the methodology implemented in this study for the statistical prediction of winter precipitation over the spatial domain of this investigation—the western United States. Figure 1 outlines the adopted framework displaying how the most recent global SST conditions from multiple temporal lags (Fig. 1a) are projected onto predictor variables varying across a range of time scales from interannual to decadal-multidecadal (Fig. 1b). The extraction of these predictor variables is based on an extended-EOF analysis of observed SSTs, which is described in the next section. Using these extended-EOF-derived predictors, a statistical model is trained and tested for the predictand of interest (Fig. 1c), which is the precipitation anomaly averaged over the winter months of November through March.

The SST predictors are obtained from an extended-EOF analysis[50] of seasonal SST anomalies in a longitudinally global domain (20°S−80°N, 0°−360°) following Nigam et al.[51] and Sengupta[52]. This analysis focuses on assessing both the spatial and temporal recurrence in pattern recognition instead of the customary focus on just the spatial structure as in a canonical EOF analysis. It yields modes of variability, which comprises the spatio-temporal SST pattern as a sequence of spatial maps (loading vector) and its related time series (principal component or PC). The additional focus on temporal recurrence provides insights into the evolutionary journey of a SST pattern or mode, i.e., its progression from the nascent phase to the mature phase and subsequently towards its decay. The primary SST analysis in this study uses a five-season-long sampling window and varimax rotation of the leading eleven global SST PCs. Additional sensitivity tests are conducted in this study (section titled 'Rationale behind the selection of multiple temporal lags') via perturbation of the primary extended-EOF analysis with variations in the sampling window length, using both one- and three-season-long sampling windows. The number of modes to be rotated and the choice for the width of the optimal seasonal sampling window are motivated by sensitivity tests reported in Nigam et al.[51] (their Section 6). This study also assessed the physicality of the SST modes via correlation analyses with NOAA fishery records to ensure that the extracted modes are realized in nature and not statistical artifacts.

On interannual time scales, four predictor modes of SST variability are derived: the canonical growth and decay phases of ENSO, Noncanonical ENSO, and Biennial variability; their mature-phase structure are shown in Supplementary Fig. 1 (top panel). A synthetic ENSO index based on these four modes is correlated with the observed Niño 3.4 SST index at 0.97; the Niño 3.4 index in both cases is defined as the area-weighted average of SST anomalies in the central-eastern equatorial Pacific over 170°W−120°W, 5°S −5°N. Additionally, this synthetic ENSO index has been documented to provide a more comprehensive representation of El Niño's influence on North American precipitation[35].

SST variability on longer time scales is captured in this analysis via two modes of decadal variability in the Pacific basin (Supplementary Fig. 1; middle panel): the Pan-Pacific (PDV-PP) and North Pacific (PDV-NP), and two multidecadal modes in the Atlantic basin (Supplementary Fig. 1; bottom panel): Atlantic multidecadal oscillation (AMO) and the low-frequency North Atlantic Oscillation (LF-NAO). The PDV-PP mode bears resemblance to the North Pacific Gyre Oscillation (NPGO[53]; correlation ~ 0.7), while the PDV-NP mode is similar to the Pacific decadal oscillation (PDO[54]; correlation ~0.8). The Atlantic modes derived from this extended-EOF analysis, specifically the AMO PC is correlated with NOAA's AMO index at 0.8 and the LF-NAO PC with Hurrell's NAO index at 0.62. The evolution of these modes of decadal-multidecadal SST variability and more interestingly, their intra-basin and inter-basin interactions are described in considerable detail in Nigam et al.[51]. In addition to modes of natural variability, this SST analysis yields a characterization of secular warming in the global oceans through the nonstationary secular trend mode (Supplementary Fig. 1).

### Construction of the seasonal prediction model based on evolving SST modes

Linear, temporally lagged, seasonal regressions of precipitation on the extracted PCs of SST variability, $RPC_i (x, y)$, in the period of analysis constitute the building blocks of the statistical prediction scheme. Next, the model utilizes the spatiotemporal structure of the observed SST anomalies across multiple, antecedent seasons (Fig. 1a). In other words, instead of focusing on the predictor field, $\psi (x, y, t_0)$, at a single snapshot of time, $t = t_0$, as customary for a canonical EOF analysis, this forecasting strategy considers antecedent sequences, or, a composite of fields staggered in time from $(t_0-n)$ to $t_0$ to generate a forecast valid at a future time, say, $t = t_0 + 1$. For example, if considering a five-season sampling of the antecedent predictor fields, the model utilizes the following composite of SST anomaly maps: [$\psi (x, y, t_0-4\Delta t); \psi (x, y, t_0-3\Delta t); \psi (x, y, t_0-2\Delta t); \psi (x, y, t_0-\Delta t); \psi (x, y, t_0)$] for building the forecast. Thereafter, this sequence of observed SST anomalies from the past seasons is projected onto the loading vectors of SST spatial patterns (Fig. 1b) identified from our extended-EOF analysis, yielding projection coefficient, $\alpha_i (t)$, for each predictor SST mode ($i = 1, 2, …, M$, where $M$ is the number of SST modes considered). Next, the time-dependent projection for each SST mode, $\alpha_i (t)$, is multiplied by their corresponding temporally invariant precipitation regression pattern, $RPC_i (x, y)$, to estimate the contribution of each SST mode toward the precipitation prediction. Finally, the overall predicted precipitation anomaly valid at time $t = t_0 + 1$ is obtained by summing up these individual SST mode-based precipitation contributions (Fig. 1c).

Please note that the study targets the extended boreal winter season spanning the months of November through March. As such, temporally lagged seasonal regressions of December−February (DJF) precipitation on the PCs of SST variability in the preceding seasons during the period of analysis constitute the building blocks for the core winter season with similar lagged regressions of monthly November and March precipitation accounting for the months in the shoulder seasons.

### Hindcast skill evaluation

Cross-validation methods are employed in this study to assess the hindcast skill of the proposed MLMS–SST model and to compare the same with that from the suite of NMME dynamical models. We employ the $n$-fold cross-validation strategy where the model is iteratively fit $n$ times, each time using

$n−1$ folds for model training and then evaluating model skill on the excluded unseen set (Supplementary Table 2). For example, when applying a seven-fold cross-validation, in the first iteration, the decade of 1949−1958 is left out and the model training only utilizes data from the remaining period of 1959−2018; the hindcast skill is then assessed over this independent decade. This process is iteratively carried forward and applied to each of the seven 10-year periods resulting in an independent model hindcast dataset.

The hindcast skill is evaluated using anomaly correlations and mean absolute error (MAE) between the observed and predicted precipitation anomalies. The statistical significance of the correlations at the 95% confidence level is assessed using the two-tailed Student's $t$-test. Subsequently, we also calculate area-averaged hindcast skill metrics (as shown in Tables 1–3 and Fig. 6) across hydrologic unit code (HUC) basins defined in the USGS watershed boundary dataset for the western U.S. The basins included in this study correspond to the HUC 2-digit level: California (divided further into northern and southern California), Pacific Northwest, Upper Colorado and Lower Colorado River basins, and the Great Basin. The HUC 2-digit basins were chosen due to their significant spatial scale, which is appropriate for seasonal forecasts considering the large-scale drivers involved at these timescales, and for their varied hydroclimate response and associated operational decision-making processes.

For skill comparison of the MLMS–SST model against the statistical model benchmark, we leverage the Mean-Square-Error Skill Score (following Murphy[55]) as follows:

$$SS = \left[ 1 - \frac{MSE(f, x)}{MSE(r, x)} \right] * 100\% \tag{2}$$

Here, MSE ($f$, $x$) is the mean-square-error of the forecast of interest ($f$) relative to observations ($x$), whereas MSE ($r$, $x$) represents that of the reference benchmark forecast. The skill score (SS) represents the percentage improvement (in terms of accuracy) in the forecast of interest ($f$), in this case, obtained from the MLMS–SST model, relative to the reference forecast ($r$), represented here as the persistence forecast.

### Data availability

All datasets used in this study are publicly available. The UK Met Office's Hadley SST data used in the extended-EOF analysis can be accessed from https://www.metoffice.gov.uk/hadobs/hadisst/. The global precipitation dataset used in model development, GPCC, is available from https://www. dwd.de/EN/ourservices/gpcc/gpcc.html, and the NOAA CPC Unified CONUS dataset from https://psl.noaa.gov/data/gridded/data.unified.daily. conus.html. NMME dynamical model hindcasts are available from the International Research Institute for Climate and Society, Columbia University's Data Library: http://iridl.ldeo.columbia.edu/SOURCES/.Models/. NMME/. The HUC basin shapefiles can be accessed from the USGS Watershed Boundary Dataset at: https://www.usgs.gov/national-hydrography/watershed-boundary-dataset. Topography data (shown in the Fig.1 schematic) is obtained from NOAA's ETOPO product, available from https://www.ngdc.noaa.gov/mgg/global/.

### Code availability

The plotting codes, written in Grid Analysis and Display System (GrADS) and the NCAR Command Language (NCL), are available upon request from the corresponding author.

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

## Acknowledgements
CW3E personnel were supported by the California Department of Water Resources Atmospheric River Program. D.E.W.'s contribution to the research was carried out at the Jet Propulsion Laboratory, California Institute of Technology, under a contract with the National Aeronautics and Space Administration. Authors A.S. and B.G. gratefully acknowledge the support from the National Aeronautics and Space Administration (Grant 80NSSC22K0926). We thank Ms. Jeanine Jones, editors Dr. Alireza Bahadori and Dr. Akintomide Akinsanola, and three anonymous reviewers for their helpful comments.

## Author contributions
A.S. and D.E.W. conceived the idea of the study. A.S. performed the observational and modeling analyses, hindcast skill assessments, generated all the figures, and wrote the manuscript. D.E.W., M.J.D., B.G., L.D.M., and F.M.R. provided feedback and contributed to the manuscript revisions.

## Competing interests
The authors declare no competing interests.
