## [Transparent Peer Review file · Communications Earth & Environment]

Role of evolving sea surface temperature modes of variability in improving seasonal precipitation forecasts

Corresponding Author: Dr Agniv Sengupta

Version 0:

Decision Letter:

Dear Dr Sengupta,

Your manuscript titled "Role of evolving sea surface temperature modes of variability in improving seasonal precipitation forecasts" has now been seen by 3 reviewers, whose comments are appended below. You will see that they find your work of some potential interest. However, they have raised quite substantial concerns that must be addressed. Specifically, you need to (1) demonstrate the added value of your model over other approaches, (2), provide a detailed explanation of the physical mechanisms governing the interaction between multi-season lagged SSTs and precipitation changes, and (3) include additional statistics to increase the robustness of the findings. In light of these comments, we cannot accept the manuscript for publication, but would be interested in considering a revised version that fully addresses these serious concerns.

We hope you will find the reviewers' comments useful as you decide how to proceed. Should additional work allow you to address these criticisms, we would be happy to look at a substantially revised manuscript. If you choose to take up this option, please either highlight all changes in the manuscript text file, or provide a list of the changes to the manuscript with your responses to the reviewers.

When resubmitting, please provide a point-by-point response to the reviewers' comments. Please submit your responses as a separate file, distinct from your cover letter where you can add responses to the Editors' comments that you do not want to be made available to the reviewers. Word files are preferred.

Important: The response to reviewers must not include any figures, tables or graphs. If you wish to respond to the reviewer reports with additional data in one of these formats, please add them to the main article or Supplementary Information, and refer to them in the rebuttal. Due to current technical limitations, any figures, tables, or graphs embedded in your rebuttal will not be included in the peer review file, if published.

If the revision process takes significantly longer than three months, we will be happy to reconsider your paper at a later date, as long as nothing similar has been accepted for publication at Communications Earth & Environment or published elsewhere in the meantime.

Please use the following link to submit your revised manuscript, point-by-point response to the reviewers' comments with a list of your changes to the manuscript text (which should be in a separate document to any cover letter), a tracked-changes version of the manuscript (as a PDF file) and any completed checklist:

Link Redacted

**** This url links to your confidential home page and associated information about manuscripts you may have submitted or be**

reviewing for us. If you wish to forward this email to co-authors, please delete the link to your homepage first **

Please do not hesitate to contact us if you have any questions or would like to discuss the required revisions further. Thank you for the opportunity to review your work.

Best regards,

Akintomide Akinsanola, PhD
Editorial Board Member
Communications Earth & Environment

Alireza Bahadori, PhD
Associate Editor
Communications Earth & Environment

EDITORIAL POLICIES AND FORMAT

If you decide to resubmit your paper, please ensure that your manuscript complies with our editorial policies and complete and upload the checklist below as a Related Manuscript file type with the revised article:

Editorial Policy Policy requirements
(Download the link to your computer as a PDF.)

- Behavioural and social science
- Ecological, evolutionary & environmental sciences
- Life sciences

<https://www.nature.com/documents/nr-reporting-summary.zip>

For your information, you can find some guidance regarding format requirements summarized on the following checklist: (<https://www.nature.com/documents/commsj-phys-style-formatting-checklist-article.pdf>) and formatting guide (<https://www.nature.com/documents/commsj-phys-style-formatting-guide-accept.pdf>).

REVIEWER COMMENTS:

Reviewer #1 (Remarks to the Author):

Sengupta et al. developed a statistical model based on the extended EOF to predict precipitation over the US in winter with a one-month lead. In my opinion, this study's statistical prediction model did not show impressive predictive skill much higher than dynamic models. Furthermore, there have been many approaches to improve the predictive skill of dynamic models, such as multi-model ensemble mean, Data post-processing of model outputs, and a combination of dynamic models and statistical methods. The author may need to demonstrate that their model has advantages relative to these approaches. The statistical method used in the study is extended EOF of SST anomalies of five consecutive seasons. This method has been used in studies of interannual variability of the East Asian monsoon and is referred to as seasonal EOF there (Wang and An 2005, GRL, <https://doi.org/10.1029/2005GL022709>). In addition, I have some detailed questions.

1) Cross-validation has been done in training the regression coefficients of precipitation on the PCs time series. However, it is still not strict cross-validation, because extended EOF of SST variability is done for the whole period of data.

2) Sensitivity tests were conducted for 1-, 3- and 5-season lags. Because the extended EOF was conducted for SST of 5 seasons, I am not sure the sensitivity tests are fair for 1- and 3-season lags.

3) Many SST variability modes have been used in this study. Though this paper focuses on predictions, the physical processes through which the SST modes influence US precipitation should be mentioned.

Reviewer #2 (Remarks to the Author):

Review report of the manuscript "Role of evolving sea surface temperature modes of variability in improving seasonal precipitation forecasts"

This study proposes a new method for predicting winter precipitation in the Western US by analyzing past seasons' sea surface temperature (SST) variations. The study demonstrates that their model performs reasonably well compared to other forecasting methods. However, I am concerned as the authors didn't explain any physical mechanisms governing the interaction between multi-season lagged SSTs and precipitation changes. Additionally, the model's skill has been low in

recent decades, suggesting other factors might influence predictions besides SST. This paper is very well written, and the topic is very relevant to society. The report needs major revision and essential clarification to be accepted for publication in this journal from my side.

Comments:

1. The author's claims that interannual to multidecadal SST variability can serve as a source of predictability to improve seasonal winter precipitation forecasts over the Western US. However, the author does not illustrate any underlying dynamical processes or mechanisms that explain how the interaction between multi past seasons SST variability leads to precipitation change in the Western US.
2. In Figure 4, aside from the period 1969-1978, I did not observe any differences in hindcast skills between the ENSO, decadal, and Atlantic modes. So, it's difficult to distinguish the impact of the decadal and Atlantic modes.
3. Figure 4: The skill is low during 2009-2018. This suggests that the model is not performing well in recent times, raising questions about the usage of the model proposed in this study.
4. The reason for the low skill in recent decades might be because the model doesn't include important factors that affect rainfall in the western US. To improve the model's skill, consider including other factors as well.
5. In Figure 5, the mean absolute error in the model is as large as the climatological mean rainfall in the Western US (especially Southern California) for different lagged predictors. It's possible that the high correlation is only because of the noise in the system.
6. Line 216: There is no change in the error from a 1-season lag to a 5-season lag, and this error map does not look like the correlation map. Correct the statement.

Reviewer #3 (Remarks to the Author):

This is a well written paper that does a nice job providing background on why this model is necessary. However, I'm still somewhat concerned this model could be overfit and have inflated skill, so hopefully some of my suggestions below help to address this concern:

(1) Unfortunately the addition of more predictors often tends to increase the skill. In order to convince the readers that is not simply happening in this paper, it would be good to do a test similar to the one outlined here:

F-test for nested linear models:

https://en.wikipedia.org/wiki/F-test#Regression_problems

The goal here is to show that the addition of an extra predictor gives a *statistically significant* better fit to the data.

(2) I'm glad to see the n-fold cross validation on line 169 and the discussion on lines 508-517. However, one thing to be mindful of is the potential pitfall explained in DelSole and Shukla (2009, citation below), who show that even cross validated models can have inflated skill because all of the data has been used to select the predictors. One might be concerned that the modes found in Figure S1 were selected based on the full set of SSTs and so are essentially screened predictors.

A simple way to convince me that is not what is happening here is to construct your model on *only* the first half of the data and then "run" the model forward on the 2nd half (test). Care should be take that none of the data in the 2nd half should be used to compute the extended EOF or any projection coefficients. In other words, I'd like to see Fig. S1 shown separately for the two halves of data and extended EOFs from the 1st half are used to make predictions for the 2nd half. Eventually, I would also do the reverse as well (build on 2nd half and then test on 1st half).

By the way, if this does in fact lower the skill show in this paper, I would still advocate for publication (provided this behavior is disclosed) b/c I think it's extremely instructive to show this happens and the authors could help others realize that is a potential pitfall when building statistical models. There is a lot of pressure to show a model has "better" skill than models that came before, so I think it's easy to overlook these issues.

DelSole, T., and J. Shukla, 2009: Artificial Skill due to Predictor Screening. *J. Climate*, 22, 331–345,

(3) I'm not at all clear why most of the analysis only goes through 2018. If the GPCP dataset is not long enough, then I would recommend using GPCP:

<https://psl.noaa.gov/data/gridded/data.gpcp.html>

Or CPC Unified gauge-based dataset for monthly means:
<https://psl.noaa.gov/data/gridded/data.cpc.globalprecip.html>

This is also a concern given that the justification of this paper is to make real time predictions. I think to establish this can be updated real time, data to the near-present should be used (through 2023-24 winter if possible).

I was particularly startled to see the analysis shown in Figure 7 (through 2021-22) since the preceding figures clearly show data only through 2018 (or 2011 in the case of NMME).

(4) Similar to my comment above, I noticed that a few of the NMME models are not current generation models and the record of evaluation stops in 2011. In particular CanCM3 is no longer updated but there are two new Canadian models: CanCM4i and GEM5-NEMO. Also, GFDL FLOR-B has been superseded by GFDL SPEAR. Since the hindcast does not go back to 1982 I would suggest subbing in another current generation model. A list of currently used models is in the legend here: <https://www.cpc.ncep.noaa.gov/products/NMME/current/images/nino34.rescaling.ENSMEAN.png>. If you don't use a current set (or the NMME average), I would avoid language like "among the top 2 performers" (line 241) and "superior to NMME" (line 391) since some of these are fairly dated models that are no longer updated. Also the full NMME system is an average of all the current generation models, which is not being assessed here.

(5) I would encourage the authors to set up a website somewhere where they update their model in real-time. If they can do this in time for publication, then it could be included as a link somewhere so interested readers can follow along with the performance in real time. Unfortunately, published statistical models that overpromise and underdeliver are a dime a dozen and that becomes quickly evident when the models are used to forecast future data. So, I think to stand out from the herd, this sort of step should be standard practice as a condition of publication. Obviously this is just a suggestion— no one will actually enforce this "rule" — but it is a good way to build trust with potential users of the product and reduce skepticism.

Minor comments:

Line 93-96: I assume the "bust" refers to a deterministic forecast. Most predictions are actually probabilistic ... the ones that come from NOAA CPC are in probabilities. So you can't look at a single (or a few) seasons as evidence of a bust since probability has meaning (e.g. 60% chance means it will happen roughly 6 in 10 times and not happen 4 in 10 times). So I would note that caveat somewhere in here.

Line 397-399: Here again, it's problematic to comment on the probabilistic outlooks by NWS b/c this type of forecast is not at all being assessed or reviewed in this paper.

Line 153: I would not look at Figure 3 and notice WY2017. It may be of recent interest but it certainly doesn't remotely rival the other peaks.

Line 159: It certainly would help if the time series was through 2023. (see major comment #3 above)

Lines 92 and 335: Jiang et al. should be mentioned, but it was clearly preceded by other people who show that the skill does not exceed 25% explained variance ($r = 0.5$). Please also reference:

Kumar, A., and M. Chen, 2020: Understanding Skill of Seasonal Mean Precipitation Prediction over California during Boreal Winter and Role of Predictability Limits. *J. Climate*, 33, 6141–6163

Communications Earth & Environment is committed to improving transparency in authorship. As part of our efforts in this direction, we are now requesting that all authors identified as 'corresponding author' create and link their Open Researcher and Contributor Identifier (ORCID) with their account on the Manuscript Tracking System prior to acceptance. ORCID helps the scientific community achieve unambiguous attribution of all scholarly contributions. You can create and link your ORCID from the home page of the Manuscript Tracking System by clicking on 'Modify my Springer Nature account' and following the instructions in the link below. Please also inform all co-authors that they can add their ORCIDs to their accounts and that they must do so prior to acceptance.

Version 1:

Decision Letter:

Dear Dr Sengupta,

Your revised manuscript titled "Role of evolving sea surface temperature modes of variability in improving seasonal precipitation forecasts" has now been seen by 3 reviewers, whose comments are appended below. You will see that the reviewers appreciate the effort you put in the revisions, but reviewer 3 continues to raise important concerns about your analysis. In light of these comments, we cannot accept the manuscript for publication, but would be interested in considering a revised version that fully addresses these serious concerns.

Specifically, for publication in Communications Earth & Environment to be appropriate, a revised manuscript must make a compelling case that the model does not suffer from overfitting and demonstrate consistent and significant model skill across the two halves of the dataset.

We hope you will find the reviewers' comments useful as you decide how to proceed. Should additional work allow you to address these criticisms, we would be happy to look at a substantially revised manuscript. If you choose to take up this option, please either highlight all changes in the manuscript text file, or provide a list of the changes to the manuscript with your responses to the reviewers.

When resubmitting, please provide a point-by-point response to the reviewers' comments. Please submit your responses as a separate file, distinct from your cover letter where you can add responses to the Editors' comments that you do not want to be made available to the reviewers. Word files are preferred. We recommend that any figures, tables or graphs that are included in the response to reviewers are also included in the main article or Supplementary Information.

If the revision process takes significantly longer than three months, we will be happy to reconsider your paper at a later date, as long as nothing similar has been accepted for publication at Communications Earth & Environment or published elsewhere in the meantime.

Please use the following link to submit your revised manuscript, point-by-point response to the reviewers' comments with a list of your changes to the manuscript text (which should be in a separate document to any cover letter), a tracked-changes version of the manuscript (as a PDF file) and any completed checklist:

Link Redacted

Please do not hesitate to contact us if you have any questions or would like to discuss the required revisions further. Thank you for the opportunity to review your work.

Best regards,

Akintomide Akinsanola, PhD
Editorial Board Member
Communications Earth & Environment

Alireza Bahadori, PhD
Associate Editor
Communications Earth & Environment

EDITORIAL POLICIES AND FORMAT

If you decide to resubmit your paper, please ensure that your manuscript complies with our editorial policies and complete and upload the checklist below as a Related Manuscript file type with the revised article:

Editorial Policy Policy requirements
(Download the link to your computer as a PDF.)

- Behavioural and social science
- Ecological, evolutionary & environmental sciences
- Life sciences

<https://www.nature.com/documents/nr-reporting-summary.zip>

For your information, you can find some guidance regarding format requirements summarized on the following checklist: (<https://www.nature.com/documents/commsj-phys-style-formatting-checklist-article.pdf>) and formatting guide (<https://www.nature.com/documents/commsj-phys-style-formatting-guide-accept.pdf>).

REVIEWER COMMENTS:

Reviewer #1 (Remarks to the Author):

The authors have solved my concerns satisfactorily. I have no further comments.

Reviewer #2 (Remarks to the Author):

I want to thank the authors for their responses and additions to the manuscript. Their point-by-point response shows that most of my comments, suggestions, and questions have been addressed. I am happy to accept it as it is.

Reviewer #3 (Remarks to the Author):

My primary concern was not the addition of current generation NMME models (my point #4) in my review. It was that I believe this model could be statistically overfit (my points #1 and #2) and I needed some additional work done to convince me that this was not the case. Now, given these responses, I believe that there is risk here that this model may suffer from overfitting.

The response to #1 is essentially that the modes are “physical and practical” so it is not necessary to evaluate whether the inclusion of each successive mode provides a *statistically significant* increase in prediction skill. Unfortunately I respectfully disagree and think that while they may have some physical basis, that doesn't eliminate the need to show each of these modes are important to the prediction. There are plenty of legitimately physical modes in climate that do not offer meaningful predictive skill.

Response to #2: I appreciate that authors splitting the data and showing the skill in Fig. S8 which is, in their view, lower. However, which figure should this skill be compared against? The most similar ones shown in the paper are split into lag times (Fig 5), so this is hard to compare. I also think this lower skill and sampling variability needs to be mentioned somewhere in the main text and not just in supplementary info, which most folks do not consider or read.

While not done here, the author should also do the training on the second half of the data and then evaluate the skill over the first half. I was curious how much the skill changes from one half to the next which is why I initially offered the suggestion. Ideally I'd like to see three comparable images: The one on the full period, Figure S8, and the opposite half of Figure S8.

If the skill changes notably among all three images, then it implies some combination of (a) these results are a result of sampling or (b) getting to my point made above, many of these modes are probably not all that important to the prediction skill. Arguably the most important prediction modes would probably be the ones that explain both a statistical significant increase in predictive skill and be robust in both halves + full record. Otherwise, it seems plausible — in fact maybe even expected — for the future data to show different levels of skill and different modes. This seems like a significant caveat.

Communications Earth & Environment is committed to improving transparency in authorship. As part of our efforts in this direction, we are now requesting that all authors identified as ‘corresponding author’ create and link their Open Researcher and Contributor Identifier (ORCID) with their account on the Manuscript Tracking System prior to acceptance. ORCID helps the scientific community achieve unambiguous attribution of all scholarly contributions. You can create and link your ORCID from the home page of the Manuscript Tracking System by clicking on ‘Modify my Springer Nature account’ and following the

instructions in the link below. Please also inform all co-authors that they can add their ORCID to their accounts and that they must do so prior to acceptance.

Version 2:

Decision Letter:

Dear Dr Sengupta,

Your manuscript titled "Role of evolving sea surface temperature modes of variability in improving seasonal precipitation forecasts" has now been seen by our reviewers, whose comments appear below. In light of their advice we are delighted to say that we are happy, in principle, to publish a suitably revised version in Communications Earth & Environment.

We therefore invite you to revise your paper one last time to comply with our format requirements and to maximise the accessibility and therefore the impact of your work.

EDITORIAL REQUESTS:

****Please take care to match our formatting and policy requirements. We will check revised manuscript and return manuscripts that do not comply. Such requests will lead to delays. ****

SUBMISSION INFORMATION:

OPEN ACCESS:

Communications Earth & Environment is a fully open access journal. Articles are made freely accessible on publication. For further information about article processing charges, open access funding, and advice and support from Nature Research, please visit <https://www.nature.com/commsenv/open-access>

Link Redacted

Best regards,

Alireza Bahadori, PhD
Associate Editor
Communications Earth & Environment

Akintomide Akinsanola, PhD
Editorial Board Member
Communications Earth & Environment

REVIEWERS' COMMENTS:

Reviewer #3 (Remarks to the Author):

I appreciate the clarification of procedures and addition of Figure S8 and have no further comments.

Dear Reviewers,

We are grateful for your insightful comments on our manuscript, which we believe have substantially improved the quality of this work. Please see our responses below (in blue) to your comments.

We summarize below the major changes made based on your comments:

- An investigation of the likely physical mechanisms governing the interactions between the multi-season, time-lagged SST modes of variability and regional precipitation changes, following suggestions from reviewers 1 and 2.
- Inclusion of additional benchmarks based on dynamical model hindcasts from the NOAA's North American Multi-Model Ensemble Phase II and related hindcast skill comparison (in line with reviewer 3's suggestions).
- Additional results highlighting the model forecasts and related verification during the most recent winters of 2017-18 to 2023-24.
- Additional analysis with the NOAA CPC Unified dataset to test any sensitivity in findings to the observational precipitation dataset used in model training.
- Performing the extended-EOF analysis of global SSTs and restricting it to the first half of the data record (until 1983) and then testing precipitation forecast skill over the second half (1984-2018), following reviewer 3's suggestion.

REVIEWER COMMENTS:

Reviewer #1 (Remarks to the Author):

Sengupta et al. developed a statistical model based on the extended EOF to predict precipitation over the US in winter with a one-month lead. In my opinion, this study's statistical prediction model did not show impressive predictive skill much higher than dynamic models. Furthermore, there have been many approaches to improve the predictive skill of dynamic models, such as multi-model ensemble mean, Data post-processing of model outputs, and a combination of dynamic models and statistical methods. The author may need to demonstrate that their model has advantages relative to these approaches.

We thank the reviewer for taking the time to review our manuscript. We would first like to point out that we already have an entire section in the Results section dedicated to skill comparison with dynamical (from the NMME Project Phases I and II) and other statistical forecasting models (lines L228-263). However, in line with the reviewer's suggestion, we have performed additional comparison of the model precipitation hindcast skill with the multi-model ensemble mean of the dynamical models analyzed as part of the study (presented as new Fig. 6 of manuscript, also shown below). Please note that additional models from NMME Phase II were incorporated into the study in line with reviewer 3's suggestion.

Fig. 6. Hindcast skill of MLMS-SST model (in red) compared with the NMME dynamical models (in red), and their multi-model ensemble mean (in black) over different constituent HUC basins in the western U.S. The dynamical models assessed here include the NCEP-CFSv2, NASA GEOS-S2S, CMC CanCM3, CanCM4i, GEM5-NEMO, GFDL FLOR-B01 and GFDL-SPEAR, with hindcasts initialized in October for the November through March winter season. The comparative assessment is performed for the common overlapping period of available hindcasts —winters 1982-83 through 2010-11 (with the exception of GFDL-SPEAR, available only from winter 1991-92). The correlations reported in each case are the area-averaged values computed over continental grid points.

The statistical method used in the study is extended EOF of SST anomalies of five consecutive seasons. This method has been used in studies of interannual variability of the East Asian monsoon and is referred to as seasonal EOF there (Wang and An 2005, GRL, <https://doi.org/10.1029/2005GL022709>).

Thank you for bringing this study to our attention. This has now been cited in the revised manuscript (L116).

In addition, I have some detailed questions.

1) Cross-validation has been done in training the regression coefficients of precipitation on the PCs time series. However, it is still not strict cross-validation, because extended EOF of SST variability is done for the whole period of data.

We appreciate the point raised by the reviewer. Below, we present two additional precipitation skill assessments over time periods where there is no overlap between SST data (utilized for model training) and forecasted precipitation.

First, we present seasonal precipitation forecasts from the MLMS-SST model and related verification for the most recent winters of 2017-18 through 2023-24 (new Fig. 8 in manuscript, also shown below). Please note that the original spatiotemporal extended-EOF analysis that yields the predictor SST modes of variability was conducted over 1900-2017. Hence, the SST analysis period is independent of the precipitation forecast period (water years 2018 through 2024). To generate the model forecast during these recent winters, we leveraged the observed SST anomalies during the five antecedent seasons to generate mode projections (time varying) and then combined them with corresponding temporally invariant regressions of precipitation on SST PCs constructed only over the historical period.

Fig. 8. Seasonal precipitation forecasts from the MLMS-SST model and related verification for the period of winters 2017-18 through 2023-24. The observational verification is based on the NOAA CPC Unified dataset. Green and brown shading denotes positive (wetter-than-normal) and negative (drier-than-normal) precipitation anomalies respectively, in units of mm/day.

From Fig. 8, we notice that the MLMS-SST model skillfully captures the observed precipitation conditions in at least five of the seven most recent winter seasons over the Western U.S., with the notable exception of winter 2022-23 which was a challenging water year for most seasonal forecast systems (see DeFlorio et al. 2024, reference #6, for details). The model forecast is characterized by accurate depictions of both below-normal (e.g., winter 2017-18, or 2020-21) and above-normal (e.g., winter 2023-24) precipitation conditions in the western U.S. We have included this new figure along with a discussion of results in the revised manuscript (L297-307).

Next, we conducted an extended EOF analysis using SST data only up to 1983, or the mid-point of the cross-validation period presented in the original analysis (following reviewer 3's suggestion), and then evaluated seasonal precipitation predictions (new Fig. S8, also shown below) over the second half of the period (1984-2018). In other words, we performed a split cross-validation where the testing period is fully independent of the training period, since no SST data from the precipitation skill assessment period (1984-2018) was used to extract the extended EOF modes that inform the model training. In fact, the model training solely relies on the historical period of 1948-1983 to learn the SST (predictor) and precipitation (predictand) relationship to generate predictions over the most recent 35 years – a tall order, given the inherently limited training data available in seasonal forecasting.

Fig. S8. Precipitation hindcast skill obtained from the MLMS-SST model over the period 1984-2018 when restricting the extended-EOF-based SST analysis to 1983. Hindcast skill is depicted here using correlations between the observed and model-predicted winter precipitation anomalies. Correlations that are statistically significant at the 95% confidence level are stippled in black. The correlation values are contoured and shaded in red at intervals of 0.1 when $\geq +0.2$, and in blue when ≤ -0.2 .

The hindcast skill assessment map still reveals notable swaths of areas in the Western U.S. with statistically significant correlations ($> +0.35$), e.g., much of Central Coast and southern California, Desert Southwest, as well as parts of the inland Pacific Northwest. We, however,

choose to retain the primary extended-EOF analysis over this approach (where the model doesn't utilize any data from the recent decades during training) for the following reasons: (i) the relatively short period used in this modified SST analysis is inadequate to skillfully extract the decadal-multidecadal modes of SST variability. The SST observed record is already fairly limited and affords less than two cycles of multidecadal variability. Hence, restricting it even further, to 1983 in this case, results in a suboptimal extraction of modes of decadal-multidecadal variability. (ii) The secular trend (related to the oceanic component of global warming) no longer emerges as the leading mode of variability when limiting SST data to only 1983. As we enter a warmer world, a robust characterization and inclusion of the influence of secular warming on regional precipitation changes will be increasingly important. We have discussed this new analysis, related findings, and limitations associated with restricting SST training data in the Supplementary material of the revised manuscript (page 11, including new Fig. S8).

2) Sensitivity tests were conducted for 1-, 3- and 5-season lags. Because the extended EOF was conducted for SST of 5 seasons, I am not sure the sensitivity tests are fair for 1- and 3-season lags.

We thank the reviewer for bringing this to our attention. On second reading, we understand that this could have been misunderstood. We would like to clarify that the sensitivity tests with 1- and 3-season lags use distinct extended-EOF analyses with one-season and three-season sampling windows respectively. We have amended the text in the Methods section to better clarify this technical detail (lines L517-520).

3) Many SST variability modes have been used in this study. Though this paper focuses on predictions, the physical processes through which the SST modes influence US precipitation should be mentioned.

Thank you for this suggestion. We have now performed additional research targeted towards elucidating the potential physical mechanisms that likely generate the SSTs' influence on regional precipitation. These findings are now presented as a new figure (Figure 9) with a dedicated section in the Results section of the revised version of the manuscript (lines L308-340). The influence is investigated via temporally lagged regressions of the lower-tropospheric streamfunction (as a measure of rotational circulation response to SST modulations) and resulting precipitation on the antecedent SST principal components (PCs), which are obtained by sampling over multiple past seasons' SST anomalies (five past seasons in this case).

Additionally, we investigate these SST influence mechanisms not just from the interannual modes of SST variability (Fig. 9a-b) but also from decadal variability modes in the Pacific (Fig. 9c-d). In this context, four modes that characterize the full spatiotemporal progression of ENSO variability were utilized, from growth to peak, and subsequently to its decay phase as well as multiple flavors – Canonical/East-Pacific and Non-canonical/Modoki ENSO.

Reviewer #2 (Remarks to the Author):

Review report of the manuscript “Role of evolving sea surface temperature modes of variability in improving seasonal precipitation forecasts”

This study proposes a new method for predicting winter precipitation in the Western US by analyzing past seasons' sea surface temperature (SST) variations. The study demonstrates that their model performs reasonably well compared to other forecasting methods. However, I am concerned as the authors didn't explain any physical mechanisms governing the interaction between multi-season lagged SSTs and precipitation changes. Additionally, the model's skill has been low in recent decades, suggesting other factors might influence predictions besides SST. This paper is very well written, and the topic is very relevant to society. The report needs major revision and essential clarification to be accepted for publication in this journal from my side. Thank you for taking the time to review our manuscript and for your constructive suggestions that are improving the article. To address the points raised by the reviewer, we have now conducted new analyses to elucidate the physical mechanisms governing the interaction between multi-season lagged SST predictor variables and regional precipitation changes. We also include additional results to address the reviewer's concerns surrounding the relatively low model skill during the recent decade. More details are provided in the responses below.

Comments:

1. The author's claims that interannual to multidecadal SST variability can serve as a source of predictability to improve seasonal winter precipitation forecasts over the Western US. However, the author does not illustrate any underlying dynamical processes or mechanisms that explain how the interaction between multi past seasons SST variability leads to precipitation change in the Western US.

The reviewer raises a good point. We have now performed additional research targeted towards elucidating the potential physical mechanisms that likely generate the SSTs' influence on regional precipitation. These findings are now presented as a new figure (Figure 9) with a dedicated section in the Results section of the revised version of the manuscript (lines L308-340). The influence is investigated via temporally lagged regressions of the lower-tropospheric streamfunction (as a measure of rotational circulation response to SST modulations) and resulting precipitation on the antecedent SST principal components (PCs), which are obtained by sampling over multiple past seasons' SST anomalies (five past seasons in this case).

Additionally, we investigate these SST influence mechanisms not just from the interannual modes of SST variability (Fig. 9a-b) but also from decadal variability modes in the Pacific (Fig. 9c-d). In this context, four modes that characterize the full spatiotemporal progression of ENSO variability were utilized, from growth to peak, and subsequently to its decay phase as well as multiple flavors – Canonical/East-Pacific and Non-canonical/Modoki ENSO.

2. In Figure 4, aside from the period 1969-1978, I did not observe any differences in hindcast skills between the ENSO, decadal, and Atlantic modes. So, it's difficult to distinguish the impact of the decadal and Atlantic modes.

To facilitate an easier understanding of the incremental value of adding additional sources of predictability, we presented the area-averaged correlations between hindcasted and observed precipitation anomalies over the different constituent hydrologic basins in the western U.S. (namely, northern and southern California, Pacific Northwest, Upper and Lower Colorado basins, and the Great Basin) in Tables 1-3. Comparing across the three tables, the highest area-averaged correlations across the six regions were concentrated in Table 2 (values highlighted in bold red), which considers contributions from ENSO variability in addition to the Pacific decadal variability modes and the secular trend. We have discussed our findings in lines L180-181 and L193-199:

“The spatial extent of higher forecast skill generally expands with an increase in the number of predictor modes in the Pacific, i.e., from $k=4$ to $k=7$ ” and “The additional consideration of decadal-multidecadal modes of variability in the Atlantic ($k=11$) results in little to no improvement in skill (Table 3) beyond that obtained from the Pacific domain ($k=7$). Our findings suggest that consideration of the combined influence of interannual modulations as well as decadal-multidecadal variability in the global oceans results in maximization of forecast skill and provides a promising new approach to improve seasonal prediction of winter precipitation across the western U.S.”

Additionally, we would like to point the reviewer to the case study presented in Figure 7 for winter 2021-22. In this case, a canonical La Niña is found to unfold upon background decadal variability signals in the Pacific over the preceding seasons leading up to the winter. We show that the additional consideration of the influence of the lower frequency modes which vary on decadal-to-multidecadal timescales in the Pacific yields a seasonal forecast (Fig. 7b) that skillfully captures the observed drier-than-normal conditions unlike the forecast based solely on ENSO (Fig. 7a).

3. Figure 4: The skill is low during 2009-2018. This suggests that the model is not performing well in recent times, raising questions about the usage of the model proposed in this study. We appreciate the reviewer’s observation. To ensure that this is not the case, we present additional results with seasonal precipitation forecasts for the most recent winters of 2017-18 through 2023-24 from the MLMS-SST model and related verification (as new Fig. 8 in the revised manuscript). We notice that the MLMS-SST model skillfully captures the observed precipitation conditions in at least five of the seven most recent winter seasons over the western U.S., with the exception of winter 2022-23 which was a challenging water year for most seasonal forecast systems (see DeFlorio et al. 2024 for details). The model forecast is characterized by accurate depictions of both below-normal (e.g., winter 2017-18, or 2021-22) and above-normal

(e.g., winter 2023-24) precipitation conditions in the western U.S. We have included this new figure along with a discussion of results in the revised manuscript (L297-307).

4. The reason for the low skill in recent decades might be because the model doesn't include important factors that affect rainfall in the western US. To improve the model's skill, consider including other factors as well.

The current study focuses on enhancements in seasonal forecast skill achieved by optimally utilizing diverse sources of SST variability evolving over multiple past seasons. While we appreciate the suggestion to include additional factors, we feel it lies beyond the scope of our current work. Having said that, we have included specific discussion on other variables that could be targeted in future research as well as possible methodologies that can best leverage multiple sources of predictive information without overfitting (lines L398-410).

“Future research could explore additional avenues to further enhance seasonal precipitation forecasting skills. One such approach might entail incorporating additional sources of predictability in the Earth system (e.g., soil moisture, terrestrial snow cover, etc.) with similar long-term memory that could be leveraged for seasonal forecasts. However, efforts to accommodate multiple sources of predictive influences and their related interactions often result in overfitting³⁸. This uncertainty can be tackled by using methods like the partial-least-square regression (PLSR), designed specifically to avoid the overfitting problem. Another option is to explore the application of machine learning algorithms, which also account for non-linearity in process interactions. However, a challenge lies with the limited size of reliable observational and reanalysis data available for training such predictive models. To overcome this limitation, hybrid strategies could be employed, such as training models in the dynamical/climate model space²⁴, but then applying transfer learning³⁹ to update model weights using real-world observations.”

5. In Figure 5, the mean absolute error in the model is as large as the climatological mean rainfall in the Western US (especially Southern California) for different lagged predictors. It's possible that the high correlation is only because of the noise in the system.

We would like to point out that Southern California is an extremely unique region exhibiting extraordinarily high variability in year-to-year precipitation, governed by the presence or absence of a relatively small number of large storms, typically landfalling atmospheric rivers (ARs), that make or break a water year (Dettinger 2016). This is evident also from Figures 2 and 3, where the winter precipitation standard deviation is $\sim 1.0\text{--}1.5 \text{ mm day}^{-1}$ against a climatological mean of $\sim 2.0 \text{ mm day}^{-1}$.

Having said that, the mean absolute errors over southern California for the figure in question are of the order of $\sim 0.5\text{--}1.0 \text{ mm day}^{-1}$, which is not as large as the climatological mean precipitation in the region. We would also like to point the reviewer to the new Figure 8 for a comparison of individual forecasts from the model and related verification for the most recent winter seasons.

This forecast application provides evidence of the potential of the model as a promising, complementary pathway to augment current operational systems in enhancing long-lead forecasting skill in the region.

Reference cited:

Dettinger, M. (2016). Historical and future relations between large storms and droughts in California. *San Francisco estuary and watershed science*, 14(2).

6. Line 216: There is no change in the error from a 1-season lag to a 5-season lag, and this error map does not look like the correlation map. Correct the statement.

On second reading, we see that the current sentence could be misunderstood. We meant to say that the bias structure and magnitude are similar across the analyzed temporal lags (Figs. 5d-f). Accordingly, we have amended the concerning sentence to: “*The MAE maps (Figs. 5d-f) exhibit a similar spatial structure and magnitude of bias across the number of analyzed temporal lags*” (L217-218).

Reviewer #3 (Remarks to the Author):

This is a well written paper that does a nice job providing background on why this model is necessary. However, I'm still somewhat concerned this model could be overfit and have inflated skill, so hopefully some of my suggestions below help to address this concern:

Thank you for taking the time to review our manuscript and for your encouraging and supportive review. We have now performed additional targeted analyses to ensure that the proposed MLMS-SST model does not suffer from inflated skill in light of the reviewer's concerns. Additionally, we have followed the reviewer's suggestions and added additional current generation NMME dynamical models. Detailed responses to the comments are presented below.

(1) Unfortunately the addition of more predictors often tends to increase the skill. In order to convince the readers that is not simply happening in this paper, it would be good to do a test similar to the one outlined here:

F-test for nested linear models:

https://en.wikipedia.org/wiki/F-test#Regression_problems

The goal here is to show that the addition of an extra predictor gives a *statistically significant* better fit to the data.

We appreciate the point raised by the reviewer. To our understanding, the F-test tells us if one regression model is better than another without requiring any of the models to be physically based. We would like to clarify that our proposed approach leverages spatially and temporally evolving global and basin-scale modes of sea surface temperature (SST) variability, where each of the predictors has a well-established physical interpretation. This is important because the SST analysis identifies optimal modes that capture variance but does not guarantee that these statistically extracted modes are realized in nature (a definitive evidence of their physical existence). Hence, we assessed the physicality of the predictor SST modes used in the current study via correlation analyses with NOAA fishery records since SST variations can have reflections in climate-sensitive marine ecosystems. We would like to refer the reviewer to Nigam et al. (2020; their Table 4) for details on these mode physicality assessments. Furthermore, we would like to point out that we have already provided evidence for statistically significant improvement in hindcast skill with the addition of decadal-multidecadal Pacific SST modes to interannual ENSO variability (Tables 1-2) as well as the practical significance of the model via skill comparison with dynamical models (Figure 6).

However, to further address the reviewer's point, we have now added additional text in the revised manuscript (lines L522-524) to clarify the physical nature of these predictor SST modes that are not simply statistical artifacts. We hope that the significance of the model is now better

articulated in the revision – not from a statistical perspective, but from physical and practical perspectives.

(2) I'm glad to see the n-fold cross validation on line 169 and the discussion on lines 508-517. However, one thing to be mindful of is the potential pitfall explained in DelSole and Shukla (2009, citation below), who show that even cross validated models can have inflated skill because all of the data has been used to select the predictors. One might be concerned that the modes found in Figure S1 were selected based on the full set of SSTs and so are essentially screened predictors.

A simple way to convince me that is not what is happening here is to construct your model on *only* the first half of the data and then “run” the model forward on the 2nd half (test). Care should be take that none of the data in the 2nd half should be used to compute the extended EOF or any projection coefficients. In other words, I'd like to see Fig. S1 shown separately for the two halves of data and extended EOFs from the 1st half are used to make predictions for the 2nd half. Eventually, I would also do the reverse as well (build on 2nd half and then test on 1st half).

By the way, if this does in fact lower the skill show in this paper, I would still advocate for publication (provided this behavior is disclosed) b/c I think it's extremely instructive to show this happens and the authors could help others realize that is a potential pitfall when building statistical models. There is a lot of pressure to show a model has “better” skill than models that came before, so I think it's easy to overlook these issues.

DelSole, T., and J. Shukla, 2009: Artificial Skill due to Predictor Screening. *J. Climate*, 22, 331–345,

Thank you for bringing this study to our attention, and we greatly appreciate the constructive suggestion that seeks to ensure that the predictive model does not suffer from inflated skill. In alignment with the reviewer's suggestion, we have now conducted an extended EOF analysis using SST data only up to 1983, i.e., the mid-point of the cross-validation period presented in the original analysis (1949-2018), and then evaluated seasonal precipitation prediction over the second half of the period (1984-2018), presented as Figure S8 below. In other words, we performed a split cross-validation here where the testing period is fully independent of the training period since no SST data from the precipitation skill assessment period (1984-2018) was used to extract the extended EOF modes that informs the model training. In fact, the model training solely relies on the historical period of 1948-1983 to learn the SST (predictor) and precipitation (predictand) relationship to generate predictions over the most recent 35 years – a tall order, given the inherently limited training data available in seasonal forecasting.

Fig. S8. Precipitation hindcast skill obtained from the MLMS-SST model over the period 1984-2018 when restricting the extended-EOF-based SST analysis to 1983. Hindcast skill is depicted here using correlations between the observed and model-predicted winter precipitation anomalies. Correlations that are statistically significant at the 95% confidence level are stippled in black. The correlation values are contoured and shaded in red at intervals of 0.1 when $\geq +0.2$, and in blue when ≤ -0.2 .

The hindcast skill assessment map still reveals notable swaths of areas in the Western U.S. with statistically significant correlations ($> +0.35$), e.g., much of Central Coast and southern California, Desert Southwest, as well as parts of the inland Pacific Northwest. However, using this suggested approach where the model doesn't utilize any data from the recent decades during training, we note an overall reduction in correlation skill which could be attributed to: (i) the relatively short period used in this modified SST analysis that is inadequate to skillfully extract the decadal-multidecadal modes of SST variability. The SST observed record is already fairly limited and affords less than two cycles of multidecadal variability. Hence, restricting it even further, to 1983 in this case, results in a suboptimal extraction of modes of decadal-multidecadal variability. (ii) The secular trend (related to the oceanic component of global warming) no longer emerges as the leading mode of variability. As we enter a warmer world, a robust characterization and inclusion of the influence of secular warming on regional precipitation changes will be increasingly important.

We have discussed this new analysis, related findings, and limitations associated with restricting SST training data in the Supplementary Material of the revised manuscript (page 11, including new Fig. S8).

(3) I'm not at all clear why most of the analysis only goes through 2018. If the GPCP dataset is not long enough, then I would recommend using GPCP:

<https://psl.noaa.gov/data/gridded/data.gpcp.html>

Or CPC Unified gauge-based dataset for monthly means:

<https://psl.noaa.gov/data/gridded/data.cpc.globalprecip.html>

This is also a concern given that the justification of this paper is to make real time predictions. I think to establish this can be updated real time, data to the near-present should be used (through 2023-24 winter if possible).

We considered the reviewer's suggestion and report below the precipitation hindcast skill using the alternative observational precipitation dataset (NOAA CPC Unified gauge-based dataset).

We choose not to include the GPCP product as it is extremely coarse (2.5° by 2.5°) for the regional hydroclimate prediction at hand, and available for a much shorter period of time (1979 to present). The model hindcast skill when trained with NOAA CPC Unified observations (Supplementary Figure S7 of the revised manuscript, also shown below) reveals little-to-no differences compared to the original GPCC-based findings (Figure 4). The GPCC version 2020 dataset was used in this study given its spatial resolution of 0.25° by 0.25° , and more importantly, its long data record spanning from January 1891 to December 2019, making it suitable for the seasonal prediction problem at hand, which is inherently data limited.

Furthermore, we preferred the GPCC dataset as it affords the opportunity to additionally predict seasonal precipitation anomalies over Mexico, particularly in areas such as Baja California where precipitation variability has been documented to be modulated by SST variability (e.g., Cavazos and Rivas 2004). Please note that the CPC-Unified product analyzed below, available from 1948 to the present, is a CONUS-only product. Meanwhile, its global counterpart product is coarser (0.5° spatial resolution) and available only since 1979, making it unsuitable for model training in the context of the present seasonal forecasting study.

Reference cited: Cavazos, T. & Rivas, D. Variability of extreme precipitation events in Tijuana, Mexico. *Clim. Res.* **25**, 229–243 (2004).

Figure S7. Hindcast skill of the MLMS–SST model across the western U.S. when performing model training with NOAA CPC Unified precipitation observations instead of GPCP version 2020. Cross-validation analyses over independent hindcast periods yield correlations between the observed and model-predicted winter precipitation anomalies. Panels a–e denote skill scores obtained when only ENSO modes of variability are considered in the predictor set, whereas subsequent panels depict skill with additional contributions from the secular trend and decadal variability modes in the Pacific (panels f–j) and in the Atlantic (panels k–o). Correlations that are statistically significant at the 95% confidence level are stippled in black. The correlation values are contoured and shaded in red at intervals of 0.1 when $\geq +0.2$, and in blue when ≤ -0.2 .

We have now included details of this additional analysis performed in light of the reviewer’s point to test any sensitivity to the observational precipitation dataset of choice (lines L471-476 of the revised manuscript).

I was particularly startled to see the analysis shown in Figure 7 (through 2021-22) since the preceding figures clearly show data only through 2018 (or 2011 in the case of NMME). We wish to clarify that for Figure 7, we used observed SST anomalies from five antecedent seasons (illustrated in Figure S5) to generate SST modal projections (time varying) and then combined them with corresponding temporally invariant precipitation regression patterns, constructed over the historical period, to generate the actual precipitation forecast (Fig. 7a-b). In other words, for the prediction, the MLMS-SST model only leverages antecedent, multi-season,

observed SST anomalies without needing to retrain the model facilitating potential real-time applications in the future. The precipitation anomalies in Fig. 7c are observed precipitation conditions, only used for verification of the MLMS model forecast.

(4) Similar to my comment above, I noticed that a few of the NMME models are not current generation models and the record of evaluation stops in 2011. In particular CanCM3 is no longer updated but there are two new Canadian models: CanCM4i and GEM5-NEMO. Also, GFDL FLOR-B has been superseded by GFDL SPEAR. Since the hindcast does not go back to 1982 I would suggest subbing in another current generation model. A list of currently used models is in the legend here:

<https://www.cpc.ncep.noaa.gov/products/NMME/current/images/nino34.rescaling.ENSMEAN.png> . If you don't use a current set (or the NMME average), I would avoid language like "among the top 2 performers" (line 241) and "superior to NMME" (line 391) since some of these are fairly dated models that are no longer updated. Also the full NMME system is an average of all the current generation models, which is not being assessed here.

Thank you for this suggestion. We have now incorporated and analyzed dynamical model hindcasts from additional current generation (NMME Phase II) models, namely, CanCM4i, GEM5-NEMO, and GFDL-SPEAR, as suggested by the reviewer. Accordingly, Figure 6 of the manuscript has been updated to highlight the hindcast skill of the MLMS-SST model and the NMME dynamical models (including the three newer models suggested by the reviewer) over different constituent hydrologic basins in the western U.S. We report no changes in the conclusions derived earlier, with the proposed MLMS-SST model still being very competitive with, or outperforming most of the NMME dynamical benchmarks across the six constituent regions of the western U.S. We have updated Table S1 to incorporate the newly analyzed models, and slightly modified the discussion of results in line with the reviewer's suggestions. Kindly note that the multi-model ensemble mean was incorporated into the study in line with reviewer 1's suggestion.

Fig. 6. Hindcast skill of MLMS-SST model (in red) compared with the NMME dynamical models (in red) and their multi-model ensemble mean (in black) over different constituent HUC basins in the western U.S. The dynamical models assessed here include the NCEP-CFSv2, NASA GEOS-S2S, CMC CanCM3, CanCM4i, GEM5-NEMO, GFDL FLOR-B01 and GFDL-SPEAR, with hindcasts initialized in October for the November through March winter season. The comparative assessment is performed for the common overlapping period of available hindcasts —winters 1982-83 through 2010-11 (with the exception of GFDL-SPEAR, available only from winter 1991-92). The correlations reported in each case are the area-averaged values computed over continental grid points.

(5) I would encourage the authors to set up a website somewhere where they update their model in real-time. If they can do this in time for publication, then it could be included as a link somewhere so interested readers can follow along with the performance in real time. Unfortunately, published statistical models that overpromise and underdeliver are a dime a dozen and that becomes quickly evident when the models are used to forecast future data. So, I think to stand out from the herd, this sort of step should be standard practice as a condition of

publication. Obviously this is just a suggestion— no one will actually enforce this “rule” — but it is a good way to build trust with potential users of the product and reduce skepticism. Thank you for this great suggestion. We wish to note that we already have a public website dedicated to subseasonal and seasonal forecast products with regular issuance of outlooks throughout the course of the winter season, typically during November through March (https://cw3e.ucsd.edu/cw3e_seasonal_outlook_archive/). One of the requirements for public issuance of these experimental outlooks is an associated peer-reviewed journal publication on the forecast methodology. Hence, we intend to transition our MLMS-SST model to this near real-time operational website shortly after publication of this work.

Minor comments:

Line 93-96: I assume the “bust” refers to a deterministic forecast. Most predictions are actually probabilistic ... the ones that come from NOAA CPC are in probabilities. So you can’t look at a single (or a few) seasons as evidence of a bust since probability has meaning (e.g. 60% chance means it will happen roughly 6 in 10 times and not happen 4 in 10 times). So I would note that caveat somewhere in here.

The reviewer raises a good point. The sentence is specific to the winter of 2015-16 when the missed precipitation forecast drew considerable public as well as research attention (please see <https://www.s2sforecasting.org/>, section titled ‘Water Year 2016’ for a comparison of the NOAA official precipitation forecast and its related verification for this winter). We have also amended the concerning sentence: “*This was evident during the 2015-16 El Niño winter when the forecasted odds favoring above-normal precipitation conditions did not materialize in the U.S. Southwest and a devastating drought continued instead ..*” (L93-95).

Line 397-399: Here again, it’s problematic to comment on the probabilistic outlooks by NWS b/c this type of forecast is not at all being assessed or reviewed in this paper.

The sentence has been changed in light of the reviewer’s concerns to just highlight the limited prediction skill of our current forecast systems in this region (related references were already presented in the Introduction section): “*The skill of existing seasonal forecasting systems is currently limited for winter precipitation in California and the Colorado River Basin, an important source of imported water supplies.*” (L446-448).

Line 153: I would not look at Figure 3 and notice WY2017. It may be of recent interest but it certainly doesn’t remotely rival the other peaks.

We agree. Thus, we have also included numerical estimates, e.g., 33% and 35% above normal in northern and southern California during WY2017 to facilitate an easier comparison of this recent wet winter with other historical winters (like WY1983).

Line 159: It certainly would help if the time series was through 2023. (see major comment #3

above)

The GPCP version 2020 dataset is used in this study due to its spatial resolution of 0.25° by 0.25° , and more importantly, its long data record spanning from January 1891 to December 2019, making it suitable for the seasonal prediction problem at hand, which is inherently data-limited. However, in line with the reviewer's suggestion, we also retrained our seasonal forecast model using NOAA CPC-Unified observations and found little-to-no sensitivity in precipitation hindcast skill, regardless of the observational precipitation dataset used.

Lines 92 and 335: Jiang et al. should be mentioned, but it was clearly preceded by other people who show that the skill does not exceed 25% explained variance ($r = 0.5$). Please also reference:

Kumar, A., and M. Chen, 2020: Understanding Skill of Seasonal Mean Precipitation Prediction over California during Boreal Winter and Role of Predictability Limits. *J. Climate*, 33, 6141–6163

Thank you for bringing this study to our attention. This paper is now cited on lines L92 and L380 as reference #29.

Our responses to reviewer comments are in blue.

REVIEWER COMMENTS:

Reviewer #1 (Remarks to the Author):

The authors have solved my concerns satisfactorily. I have no further comments.

Thank you for taking the time to review our manuscript and for your supportive review.

Reviewer #2 (Remarks to the Author):

I want to thank the authors for their responses and additions to the manuscript. Their point-by-point response shows that most of my comments, suggestions, and questions have been addressed. I am happy to accept it as it is.

We thank the reviewer for taking the time to review our manuscript and for offering helpful suggestions that have improved the manuscript.

Reviewer #3 (Remarks to the Author):

My primary concern was not the addition of current generation NMME models (my point #4) in my review. It was that I believe this model could be statistically overfit (my points #1 and #2) and I needed some additional work done to convince me that this was not the case. Now, given these responses, I believe that there is risk here that this model may suffer from overfitting.

The response to #1 is essentially that the modes are “physical and practical” so it is not necessary to evaluate whether the inclusion of each successive mode provides a *statistically significant* increase in prediction skill. Unfortunately I respectfully disagree and think that while they may have some physical basis, that doesn’t eliminate the need to show each of these modes are important to the prediction. There are plenty of legitimately physical modes in climate that do not offer meaningful predictive skill.

We thank the reviewer for taking the time to review the revised manuscript. Our study focuses on leveraging the oceanic memory of SST predictors evolving over multiple past seasons to improve seasonal precipitation forecasts over the western U.S. In response to the reviewer’s concern, we would like to clarify that we are not suggesting that it is imperative to include each of the successive modes extracted from our spatiotemporal SST analysis in the context of precipitation prediction. Rather, we aim to highlight the incremental value of incorporating specific sources of predictability—ranging from interannual to decadal-multidecadal timescales—across different hydrologic basins of the western U.S. (northern and southern California, Pacific Northwest, Upper and Lower Colorado basins, and the Great Basin), as shown in Tables 1-3. Our findings demonstrate a statistically significant improvement in correlation skill (cf. Table 2) with the inclusion of Pacific decadal variability modes to the base scheme that only considers ENSO variability (Table 1). However, further inclusion of decadal-multidecadal modes of variability from the Atlantic basin provides little to no additional improvement in skill (Table 3) beyond that obtained from the Pacific domain.

We would also like to refer the reviewer to the case study presented in Figure 7 for winter 2021-22. In this case, a canonical La Niña is found to unfold upon background decadal variability signals in the Pacific over the preceding seasons leading up to the winter. We show that the additional consideration of the influence of the lower frequency modes varying on decadal-to-multidecadal timescales in the Pacific yields a seasonal forecast (Fig. 7b) that skillfully captures the observed drier-than-normal conditions, unlike the forecast based solely on ENSO variability (Fig. 7a).

Response to #2: I appreciate that authors splitting the data and showing the skill in Fig. S8 which is, in their view, lower. However, which figure should this skill be compared against? The most similar ones shown in the paper are split into lag times (Fig 5), so this is hard to compare. I also think this lower skill and sampling variability needs to be mentioned somewhere in the main text

and not just in supplementary info, which most folks do not consider or read.

While not done here, the author should also do the training on the second half of the data and then evaluate the skill over the first half. I was curious how much the skill changes from one half to the next which is why I initially offered the suggestion. Ideally I'd like to see three comparable images: The one on the full period, Figure S8, and the opposite half of Figure S8.

If the skill changes notably among all three images, then it implies some combination of (a) these results are a result of sampling or (b) getting to my point made above, many of these modes are probably not all that important to the prediction skill. Arguably the most important prediction modes would probably be the ones that explain both a statistically significant increase in predictive skill and be robust in both halves + full record. Otherwise, it seems plausible — in fact maybe even expected — for the future data to show different levels of skill and different modes. This seems like a significant caveat.

We would like to thank the reviewer for their additional suggestions intended to enhance the robustness of our study. In the prior revision, we conducted extended-EOF analysis using SST data only up to 1983, i.e., the mid-point of the period of analysis (1949-2018), performed model training using predictors extracted only from the first half, and then evaluated seasonal precipitation prediction skill over the independent second half of the period (1984-2018). Following the reviewer's suggestion, we now report the model skill by reversing the model training and testing periods (i.e., training on the second half of the record and testing on the first half). In other words, we performed split cross-validation for the aforementioned analyses where the testing period is fully independent of the training period, as no SST data from the precipitation skill assessment period are used to extract the extended-EOF modes that inform the model training. Please note that the predictors used are the combination of ENSO variability and Pacific decadal variability modes, i.e., the scheme that provides a statistically significant increase in prediction skill above the ENSO-only base scheme, as reported in the response above.

However, we would like to note in this context that this strategy of training on one-half and testing on the other half assumes stationarity between the two halves of the data record, which may not be a valid assumption. Furthermore, seasonal forecasting is an inherently data-limited problem, where the amount of reliable observational data available for model training is of the order of a few seasons per year. Given that the analyses requested further exclude half of the available data, the constraints on training are even more pronounced. Nevertheless, we see the value in the reviewer's suggestion and therefore conducted these additional analyses in alignment with this request.

Fig. S8. Additional precipitation hindcast skill assessments of the MLMS-SST model (a) performed over the full period of analysis, **(b)** by training on the first half and testing on the independent second half of the record, and **(c)** by training on the second half and testing on the independent first half of the record. The primary extended-EOF analysis is modified in (b) and (c) to leverage predictors extracted solely from the corresponding half of the data. Hindcast skill is depicted here using correlations between the observed and model-predicted winter precipitation anomalies. Correlations that are statistically significant at the 95% confidence level are stippled in black. The correlation values are contoured and shaded in red at intervals of 0.1 when $\geq +0.2$, and in blue when ≤ -0.2 .

The results, now presented as new Fig. S8 of the revised manuscript, show no notable changes in the precipitation hindcast skill across the three periods of assessment requested by the reviewer, quelling concerns related to elevated skill due to sampling variability. The greatest coverage of areas with statistically significant correlation skill remain over the Pacific Northwest, California, the Lower Colorado River basin, and parts of the Great Basin (Fig. S8a-c). We also include specific discussion lines related to these additional precipitation skill assessments in the main text (L397-401), as recommended by the reviewer:

“Additional precipitation hindcast skill assessments are conducted to ensure the model does not exhibit elevated skill due to sampling variability. These analyses utilizing predictors extracted from one-half of the analysis period for model training and testing on the other half (and vice versa) reveal no notable differences in realized skill (Supplementary Fig. S8).”

Additionally, we have also incorporated a dedicated section (revising the previous one) to discuss these new analyses, related findings, and caveats on page 11 of the Supplementary Material:

“In the main article, we investigated the hindcast skill of the MLMS–SST model over the western United States during the boreal winter (November to March) using a n -fold cross-validation

approach and reported skill over the recent five decades ranging from 1969 to 2018. Additionally, here we perform split cross-validation analyses by modifying the primary extended-EOF analysis to use SST data only for one-half of the data record and then evaluating seasonal precipitation forecasts over the independent second half, presented as Supplementary Figure S8. In other words, no SST data from the independent precipitation skill assessment period is used to extract the extended-EOF modes that inform the model training. The model training solely relies on one-half of the period of analysis, i.e., either 1949-1983 or, 1984-2018, to learn the SST modes (predictors) and precipitation (predictand) relationship. The motivation for these analyses was to ensure that the model does not exhibit elevated skill due to sampling variability.

The hindcast skill assessment maps reveal similar swaths of areas in the western U.S. with statistically significant correlations across the three assessment periods, e.g., over the Pacific Northwest, California, Lower Colorado River basin, and parts of the Great Basin. However, it is important to note a couple of caveats in this modified approach, where the model training does not utilize half of the available data for seasonal forecasting, which is an inherently data-limited problem. First, the relatively short period of the modified SST analysis may be inadequate to skillfully extract the decadal-multidecadal modes of SST variability. The SST observed record is already fairly limited and affords less than two cycles of multidecadal variability. Hence, restricting it even further may lead to a suboptimal extraction of modes of decadal-multidecadal variability. Second, in the curtailed analysis (e.g., when utilizing training data only up to 1983), the secular trend (related to the oceanic component of global warming) might not emerge as the leading mode of variability as in the case of primary extended-EOF analysis that leverages the full data record. As we enter a warmer world, a robust characterization and inclusion of the influence of secular warming on regional precipitation changes will be increasingly important.”

To summarize, in response to the reviewer’s suggestions, we have conducted additional split cross-validation analyses, complementing the previously reported n -fold cross-validation. We hope that these new analyses provide the journal’s readership with further evidence of the robustness of our statistical model. We are grateful to the reviewer for the insightful feedback and hope that these additional results sufficiently address concerns related to potential overfitting.

Our responses to reviewer comments are in blue.

REVIEWER COMMENTS:

Reviewer #3 (Remarks to the Author):

I appreciate the clarification of procedures and addition of Figure S8 and have no further comments.

We thank the reviewer for the constructive suggestions that have improved the quality of the manuscript.